# Molecular principles underlying dual RNA specificity in the *Drosophila* SNF protein

Gert Weber [1,2], Gregory T. DeKoster[3], Nicole Holton[1], Kathleen B. Hall[3] & Markus C. Wahl [1,2]

The first RNA recognition motif of the *Drosophila* SNF protein is an example of an RNA binding protein with multi-specificity. It binds different RNA hairpin loops in spliceosomal U1 or U2 small nuclear RNAs, and only in the latter case requires the auxiliary U2A′ protein. Here we investigate its functions by crystal structures of SNF alone and bound to U1 stem-loop II, U2A′ or U2 stem-loop IV and U2A′, SNF dynamics from NMR spectroscopy, and structure-guided mutagenesis in binding studies. We find that different loop-closing base pairs and a nucleotide exchange at the tips of the loops contribute to differential SNF affinity for the RNAs. U2A′ immobilizes SNF and RNA residues to restore U2 stem-loop IV binding affinity, while U1 stem-loop II binding does not require such adjustments. Our findings show how U2A′ can modulate RNA specificity of SNF without changing SNF conformation or relying on direct RNA contacts.

[1] Laboratory of Structural Biochemistry, Freie Universität Berlin, Takustraße 6, D-14195 Berlin, Germany. [2] Helmholtz-Zentrum Berlin für Materialien und Energie, Macromolecular Crystallography, Albert-Einstein-Straße 15, D-12489 Berlin, Germany. [3] Department of Biochemistry and Molecular Biophysics, Washington University Medical School, St. Louis, Missouri 63110, USA. Correspondence and requests for materials should be addressed to G.W. (email: gweber@posteo.de) or to K.B.H. (email: kathleenhal@gmail.com) or to M.C.W. (email: mwahl@zedat.fu-berlin.de)

RNA-binding proteins (RBPs) often employ folded domains to specifically recognize their target RNAs[1, 2]. Transcriptome-wide RNA binding studies[3–5] have revealed hundreds of RBPs bound to diverse RNA molecules. Many RBPs can bind to multiple RNA targets that lack obvious or strict consensus sequences[6], suggesting that their RNA affinities and specificities can be modulated in the molecular context in which the recognition events take place. However, only few such examples have been analyzed in detail, leaving the molecular mechanisms that regulate RNA affinities and specificities of such RBPs in many cases unclear.

The U1A/U2B″/SNF family of RBPs is found in the U1 and/or U2 small nuclear ribonucleoprotein (snRNP) components of the spliceosome[7–9]. Jawed vertebrates use U1A to bind stem-loop II of U1 snRNA (U1$^{SLII}$) and U2B″ in conjunction with the leucine-rich repeat (LRR) protein U2A′ to bind SLIV of U2 snRNA (U2$^{SLIV}$)[10]. Both U1A and U2B″ employ an N-terminal RNA recognition motif (RRM) to specifically recognize their RNA targets. RRMs are the most widespread type of RNA-binding domain in eukaryotes[2]. While different RRMs can interact with target RNAs in diverse ways[11], crystal structures of human (h) U1A RRM1 (hU1A$^{RRM1}$) alone[12] and in complex with human U1 snRNA stem-loop II (hU1$^{SLII}$)[13] have revealed the canonical mode of how RRMs bind single-stranded RNA sequences via the conserved RNP1 and 2 motifs on the exposed surface of a β-sheet. Moreover, structural analysis of human U2B″ RRM1 (hU2B″$^{RRM1}$) bound to human U2 snRNA stem-loop IV (hU2$^{SLIV}$) in context of the human U2A′ (hU2A′) protein[14] have shown how the two proteins together provide a binding surface for the stem of hU2$^{SLIV}$, and how exchange of key residues leads to altered RNA loop specificities in hU1A$^{RRM1}$ and hU2B″$^{RRM1}$.

While segregation of the U1A$^{RRM1}$ and U2B″$^{RRM1}$ RNA-binding specificities and selective binding of U2B″$^{RRM1}$ to U2A′ can be understood based on their divergent sequences in jawed vertebrates[15], other metazoan species use SNF to bind both U1$^{SLII}$ and U2$^{SLIV}$[10], only in the latter case in conjunction with U2A′. For example, *Drosophila melanogaster* (d) SNF RRM1 (dSNF$^{RRM1}$) binds *Drosophila* U1$^{SLII}$ (dU1$^{SLII}$) with high affinity in vitro, but its affinity for *Drosophila* U2$^{SLIV}$ (dU2$^{SLIV}$) is up to 100-fold weaker[16], depending on the conditions. Analysis of binding thermodynamics revealed that *Drosophila* U2A′ (dU2A′) cooperatively restores high-affinity binding of dSNF$^{RRM1}$ to dU2$^{SLIV}$, under conditions where it does not influence dSNF$^{RRM1}$ binding to dU1$^{SLII}$[17], although it can also associate with the latter complex[16]. Consistently, the ternary complex is found only in the U2 snRNP in vivo[18]. Until now the mechanistic principles that allow dSNF$^{RRM1}$ to bind dU1$^{SLII}$ directly, yet be modulated by dU2A′ to selectively enhance its affinity for dU2$^{SLIV}$, remain unclear.

Here, we report crystal structures of four states of dSNF$^{RRM1}$ at 2 Å resolution or better: the free dSNF$^{RRM1}$ (residues 1–96; dSNF$^{1–96}$), dSNF$^{1–96}$–dU1$^{SLII}$, dU2A′–dSNF$^{1–96}$, and dU2A′–dSNF$^{1–96}$–dU2$^{SLIV}$. Altogether with the characterization of dSNF$^{RRM1}$ dynamics alone and in complexes by NMR spectroscopy and targeted mutagenesis combined with binding studies, we delineated how the intrinsic dSNF$^{RRM1}$ RNA-binding capacity and specificity are fine-tuned by networks of intra-molecular interactions that modulate dSNF$^{RRM1}$ dynamics, and how the auxiliary dU2A′ protein can capitalize on intrinsic dSNF$^{RRM1}$ flexibility to gear dSNF$^{RRM1}$ binding specifically to dU2$^{SLIV}$.

## Results

**dSNF$^{RRM1}$ exhibits high intrinsic structural flexibility.** The sequence of dSNF$^{RRM1}$ is 84% identical to hU1A$^{RRM1}$ and hU2B″$^{RRM1}$ and includes the amino acid triad Tyr-Gln-Phe that defines this family (Fig. 1a). We determined a crystal structure of dSNF$^{RRM1}$ based on a construct containing residues 1–96 of dSNF (dSNF$^{1–96}$) at 1.49 Å resolution (Supplementary Table 1), showing that the protein adopts a classic RRM fold, with Tyr10-Gln51-Phe53 displayed on the surface of its four-stranded anti-parallel β-sheet (Fig. 1b), in agreement with the previous NMR structure of dSNF$^{RRM1}$ [19]. Comparison of the six crystallographically independent dSNF$^{1–96}$ molecules indicated intrinsic conformational flexibility in the RNA-binding surface of dSNF$^{1–96}$, predominantly affecting loop L3 (residues 42–51; Cα root-mean-square deviation (rmsd) 0.16–1.10 Å with a maximum displacement of 1.47 Å) and to a lesser extent L1 (residues 13–19; rmsd 0.10–0.14 Å with a maximum displacement of 0.21 Å), as well as side chain conformations in the Tyr-Gln-Phe motif (Fig. 1b, c). Furthermore, the conformations and positions of the N-terminal extensions (residues 1–7), which in one case forms an additional α-helix, and the C-terminal loops L6 (residues 84–87) and α3 helices (residues 88–94) differ among the six dSNF$^{1–96}$ copies, in agreement with solution NMR data showing that the latter element undergoes extensive ps-ns motions[20].

Crystal packing will limit the authentic conformational ensemble that the protein exhibits in solution. We therefore carried out NMR experiments to assess dSNF$^{RRM1}$ backbone amide dynamics in solution. For these experiments we used two dSNF$^{RRM1}$ constructs, dSNF$^{1–96}$ and dSNF$^{1–101}$. Both short and long versions of dSNF$^{RRM1}$ showed intermediate dynamics in most elements of the free protein (Fig. 2a, d). $^{15}$N/$^{1}$H $\Delta R_2$ experiments ($\Delta R_2$—enhancement of the intrinsic $R_2$ due to conformational exchange; here we measure $\Delta R_{2,eff} = R_{2,app}(\nu_{CPMG}50\ Hz) - R_{2,app}(\nu_{CPMG}1000\ Hz)$) at 23 °C revealed motions on the μs-ms timescale affecting dSNF$^{RRM1}$ residues along its entire length. Helix α1 (residues 20–34) is particularly mobile, indicated by large $\Delta R_{2,eff}$ components ($\Delta R_{2,eff} = 20\ s^{-1}$ [50 ms]; Fig. 2a). Some L5 residues (70–76) also exhibit dynamics >10 s$^{-1}$ (100 ms). Truncating α3 at residue 96 causes the C-terminal region of the corresponding construct to become more flexible on the ns-ps and ms timescales as the helical structure is disrupted (Fig. 2a and see below), consistent with three copies of dSNF$^{1–96}$ in the crystal structure entirely lacking electron density for the α3 region. We did not obtain $\Delta R_{2,eff}$ values for several L3 residues because of line broadening or absence of a resonance (orange labels in Fig. 2a–c), indicating intermediate conformational exchange on this timescale or exchange with solvent protons. Only α2 residues exhibited no measurable exchange. Altogether these analyses document significant conformational flexibility throughout virtually the entire dSNF$^{RRM1}$, peaking at the termini, α1, L3, and L5.

**Structure of a dSNF$^{1–96}$–dU1$^{SLII}$ complex.** We used a 22-residue, blunt-ended dU1$^{SLII}$ construct to determine the crystal structure of a dSNF$^{1–96}$-dU1$^{SLII}$ complex at 2.0 Å resolution (Supplementary Table 1). dU1$^{SLII}$ contains a 10-nucleotide (nt) loop (A7-C16) closed by a C6:G17 Watson-Crick base pair (Fig. 3a; to simplify comparisons between the various RNAs, we numbered them identically according to the provided scheme). The center of the loop arches across the dSNF$^{1–96}$ β-sheet, L3 protrudes through the RNA loop, while L1, L6, and α3 border the outside of the loop (Fig. 3b). The overall structure closely resembles the previously analyzed hU1A$^{RRM1}$–hU1$^{SLII}$ complex[13] (Fig. 3b).

The similar positioning of the U1$^{SLII}$ hairpins on dSNF$^{RRM1}$/hU1A$^{RRM1}$ is reflected in similar protein contacts to the 5′ branches of the RNA stems (Fig. 3c). However, molecular interactions in other parts of the complexes differ in detail. In the human system, the side chains of Ser46 and Ser48 together with backbone amides of Arg47 and Lys50 (L3) maintain an extensive, water-mediated hydrogen bonding network with nucleotides A12-G17 (Fig. 3d, bottom panel). In contrast in the *Drosophila* complex, the corresponding dSNF$^{RRM1}$ residues (Leu43, Thr45,

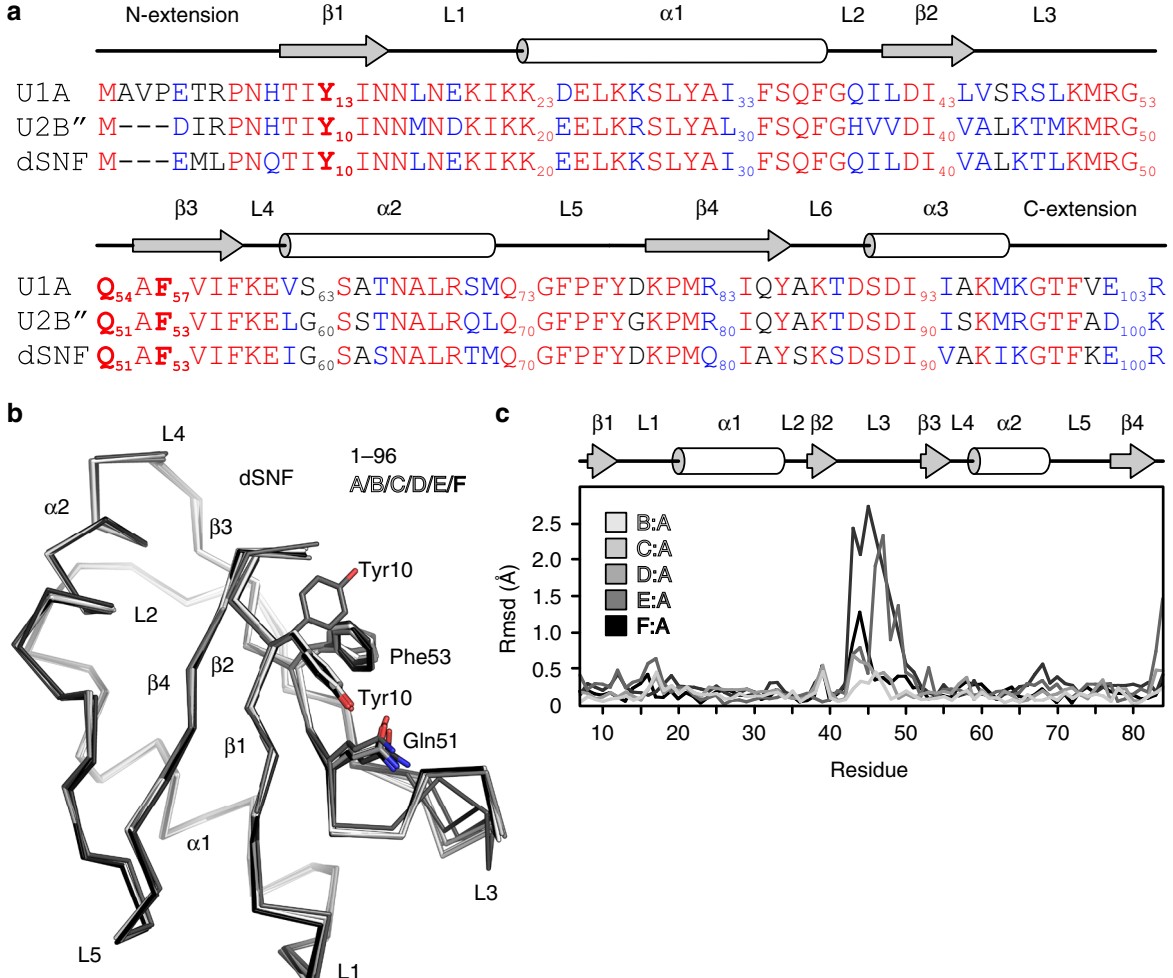

**Fig. 1** dSNF$^{1-96}$ structure and conformational flexibility. **a** Sequence alignment of dSNF$^{RRM1}$, hU1A$^{RRM1}$ and hU2B"R$^{RM1}$. Identical residues—red; conserved residues —blue. The Tyr10-Gln51-Phe53 triad is highlighted in bold. Secondary structure elements as observed in a dSNF$^{1-96}$ crystal structure are shown above the alignment (a very short β-sheet within L5 has been omitted for clarity). **b** Overlay of the six dSNF$^{1-96}$ molecules (A-F) in an asymmetric unit of a dSNF$^{1-96}$ crystal (different shades of gray). Conserved RNA-binding residues (Tyr10-Gln51-Phe53) are shown as sticks. **c** Cα rmsd's of crystallographically independent dSNF$^{1-96}$ copies relative to molecule A. The N- and C-terminal extensions and the L6 and α3 regions adopt very different conformations in the different dSNF$^{1-96}$ copies and have been omitted from the comparison

Lys44, and Lys47, respectively) do not allow the formation of a similarly extensive water network. While Thr45, Lys44, and Lys47 engage in water-mediated contacts to the phosphate oxygens of C16 and G17, a more expanded network is prevented by Leu43, which replaces Ser46 of hU1A$^{RRM1}$ in dSNF$^{RRM1}$ (Fig. 3d, top and middle panels). As a consequence, C14 can adopt two alternative conformations. In one conformation, it is bulged out and does not contact dSNF$^{RRM1}$ (two of the three crystallographically independent complexes in the crystal; Fig. 3d, top panel); in the other, its base is flipped towards dSNF$^{RRM1}$ (Fig. 3d, middle panel), where it rests on a hydrophobic surface formed by Val41 (β2) and Leu43 (L3) and where it engages in hydrogen bonds to the side chain of Lys24 (α1) and the backbone carbonyl group of Ile40 (β2; one of the crystallographically independent complexes; Fig. 3e). The corresponding U14 in the human system is disordered in two of the observed complexes (PDB ID 1URN)[13]; in the third, it is stabilized above C15 by water-mediated interactions to the phosphate of C13 and a direct contact to the base of C16 (Fig. 3d, bottom panel). Lys47 (L3) contacts the phosphate on the 5′ side of A12 in the *Drosophila*

complex (Fig. 3d, top and middle panels), while the equivalent hU1A$^{RRM1}$ Lys50 does not (Fig. 3d, bottom panel).

Several residues on the RRM β-strand surface and in the C-terminal extension of the RRM are unique to dSNF and lead to different contacts to the RNA loops in *Drosophila* and human (Fig. 3d). While Asn12 (Asn15 in hU1A$^{RRM1}$) hydrogen bonds to N7 of G10 in both organisms, it is stabilized by Gln80 (Arg83 in hU1A$^{RRM1}$) only in *Drosophila*. Gln85 of hU1A$^{RRM1}$ hydrogen bonds to the C11 4-amino group, while the corresponding Ala82 in *Drosophila* does not allow for a similar interaction. Conversely, Ser84 in dSNF$^{RRM1}$ engages in a hydrogen bond to the 6-amino group of A12, while the equivalent Ala88 of hU1A$^{RRM1}$ cannot. Instead, the 6-amino group of A12 and the 4-amino group of C13 are jointly bound by Thr89 in hU1A$^{RRM1}$, while the equivalent Ser86 does not engage in such interactions in *Drosophila*. Despite these differences in detail, our structural data are consistent with similar patterns of recognition by dSNF$^{RRM1}$ and hU1A$^{RRM1}$ for their respective U1$^{SLII2I}$.

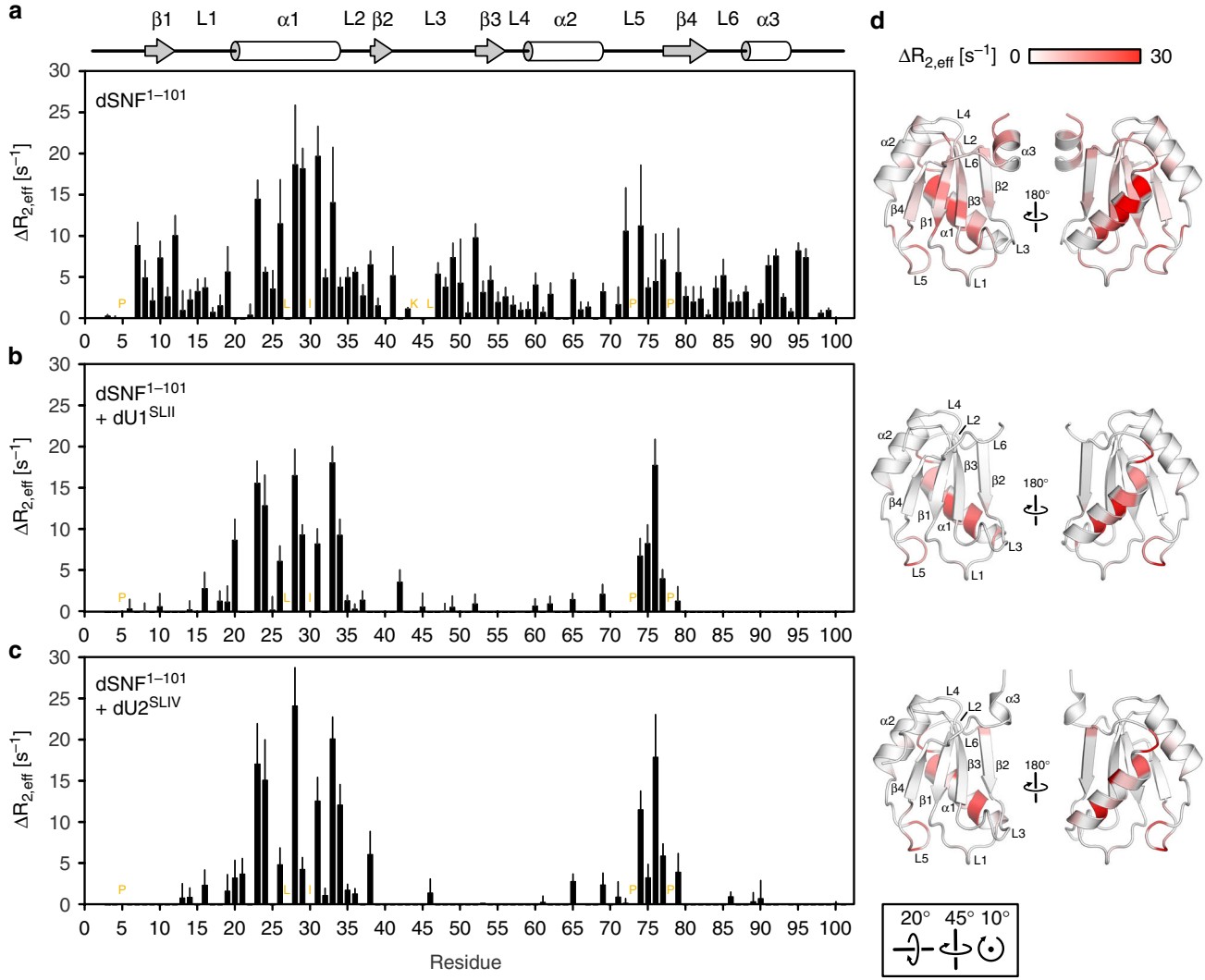

**Fig. 2** dSNF[1–101] 15N/1H backbone dynamics change upon RNA binding. **a–c** Values of $\Delta R_{2,eff}$, calculated from end points of a CPMG experiment; $\Delta R_{2,eff} = R_{2,app}(\nu_{CPMG}50 \text{ Hz})–R_{2,app}(\nu_{CPMG}1000 \text{ Hz})$. Positions of secondary structure elements are shown on the top. **a** dSNF[1–101], **b** dSNF[1–101] bound to dU1[SLII], **c** dSNF[1–101] bound to dU2[SLIV]. Errors were determined from the propagation of base plane rms noise. In **a**, **b** and **c**, amides from residues L27 and I30 have no signal from exchange broadening, while in (**a**), amides from K44 and L46 are exchange-broadened and not quantifiable (orange labels). Prolines (orange labels) do not yield a signal. 50 mM KCl, 20 mM sodium cacodylate, pH 6.5, in 90% 1H2O, 10% 2H2O; 23 °C; 700 MHz. **d** Orthogonal views of $\Delta R_{2,eff}$ values mapped onto the structure of dSNF[1–96] in isolation (top), bound to dU1[SLII] (middle) and bound to dU2A′ and dU2[SLIV] (bottom). A scale bar is shown on the top. The orientation of the left panels relative to Fig. 1b is shown at the bottom

**dU2[SLIV] binds in a similar fashion to dSNF[1–101] as dU1[SLII].** NMR chemical shift perturbations of dSNF[1–101] backbone amides upon dU1[SLII] binding[22] agree well with the crystal structure (Fig. 4a, d). L3 and the C-terminal region experience the largest changes between free and dU1[SLII]-bound dSNF[1–101], consistent with L3 and L6 directly contacting the RNA loop. Additional perturbations are seen in β1, L1 and β4, in agreement with direct contacts of L1 to the RNA stem (Lys19-G2/C3; Figs. 3c) and β4 binding the RNA loop (Lys77-U9), as well as Gln80–Asn12 interactions that are stabilized when RNA is bound (Fig. 3d, top and middle panels). Little chemical shift perturbation was observed in α1, α2, and L5 that form the backside of dSNF[1–101] and do not directly engage the RNA ligand.

dSNF[RRM1] binds dU2[SLIV] with 10 to 100-fold weaker affinity compared to dU1[SLII], depending on the salt concentration, thus hampering crystallization. However, super-stoichiometric amounts of dU2[SLIV] ([dSNF[1–101]]:[dU2[SLIV]] 1:4) allowed analysis of the dSNF[1–101]–dU2[SLIV] complex by solution NMR. The pattern of chemical shift perturbations following dU2[SLIV] binding to dSNF[1–101] resembles that seen for dSNF[1–101]–dU1[SLII] (Fig. 4b–d), indicating a similar overall binding mode, but exhibits notable differences in detail. Chemical shift differences are most extensive in the β2-L3-β3 and C-terminal regions (Fig. 4c, d). The most prominent differences are seen for Leu46 (L3), whose backbone amide hydrogen bonds to the phosphate of G17 and whose side chain stacks on the G17 base of the loop-closing base pair in the dSNF[1–96]–dU1[SLII] complex, and Asp89 (α3) that caps C13 in the dSNF[1–96]–dU1[SLII] complex. Notably, in dU2[SLIV] a U6:G17 wobble pair replaces the Watson-Crick C6:G17 loop-closing base pair of dU1[SLII], and C13 in the loop of dU1[SLII] is replaced by G13 in dU2[SLIV] (Fig. 3a). While chemical shift differences in Asp89 may thus be explained by similar interactions with different types of residues in dU1[SLII] and dU2[SLIV], chemical shift differences in Leu46 suggest that dSNF[RRM1] differentially recognizes the different configurations of the loop-closing base pairs in the two RNAs.

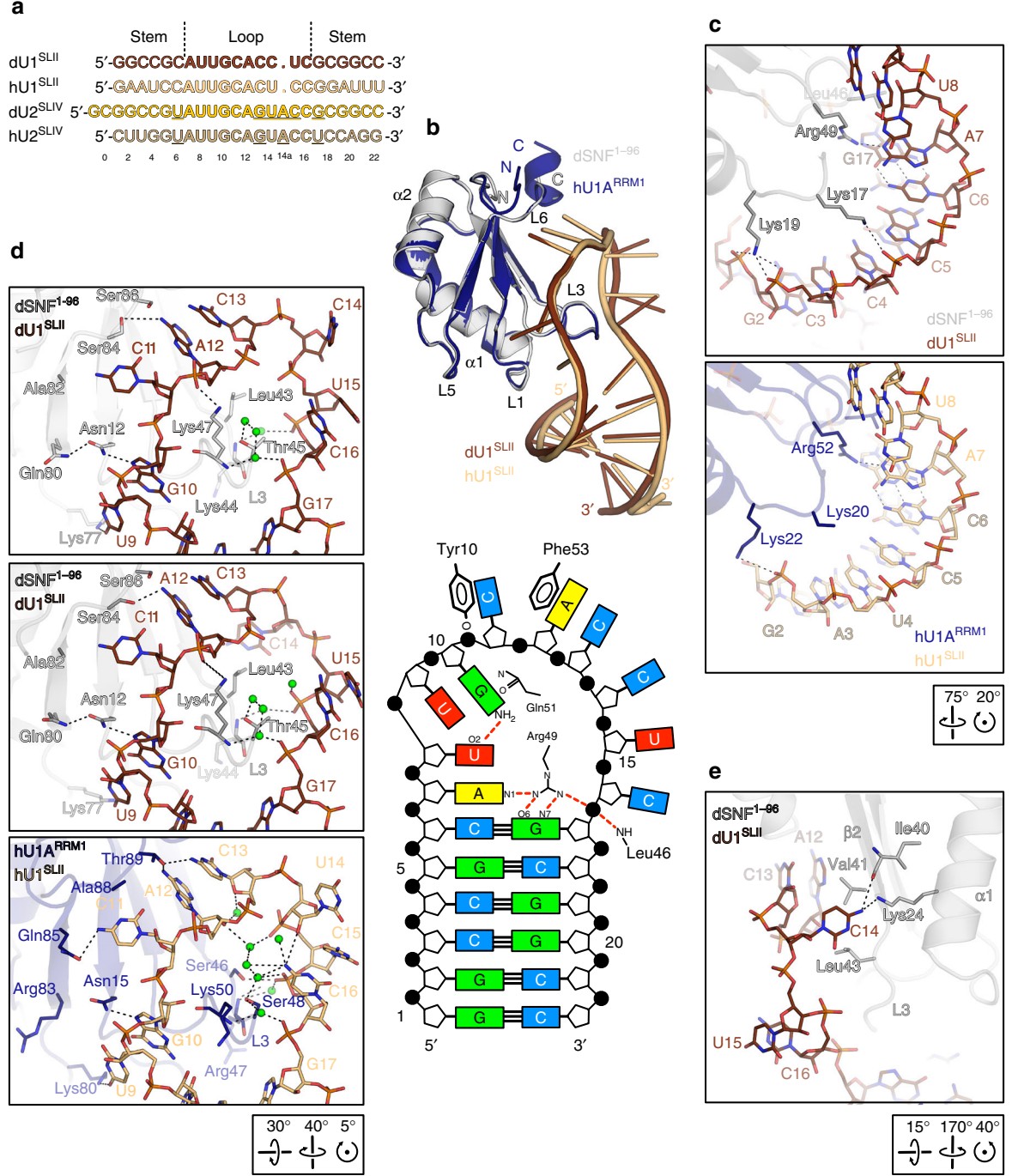

**Fig. 3** dSNF[1–96]–dU1[SLII] structure compared to hU1A[RRM1]–hU1[SLII]. **a** Alignment of dU1[SLII] (brown), hU1[SLII] (light orange), dU2[SLIV] (gold) and hU2[SLIV] (light brown) RNAs (*Drosophila* RNAs as used in the present study; human RNAs as used in refs. [13, 14]). Loop residues are in bold. Sequence differences in the loops and the loop-closing base pairs of U2[SLIV] compared to U1[SLII] in the *Drosophila* and human systems are underlined. **b** Top: hU1A[RRM1]–hU1[SLII] complex (dark blue/light orange; PDB ID 1URN, chains B and Q[13]) superimposed on the crystal structure of a dSNF[1–96]–dU1[SLII] complex (gray/brown) according to the hU1A[RRM1]/dSNF[1–96] subunits. Orientation of dSNF[1–96] as in Fig. 1b. Bottom: Scheme of selected protein-RNA interactions in the dSNF[1–96]-dU1[SLII] complex. **c** dSNF[1–96] contacts to the stem of dU1[SLII] (top panel) and comparison to the human system (bottom panel). Lys20 Cδ, Cε and Nε coordinates are not contained in the human structure. **d** dSNF[1–96] contacts to the dU1[SLII] loop and comparison to the human system. **e** C14 of dU1[SLII] flipped towards dSNF[1–96] in one of the complexes in the crystal. In this and the following figures: Stick representations are colored by atom type; carbon—color of the respective molecule; nitrogen—blue, oxygen—red, phosphorus—orange, sulfur—yellow; water oxygens are shown as green spheres; dashed lines represent hydrogen bonds or salt bridges. Orientations relative to Fig. 1b are indicated by boxed rotation symbols

**dSNF[1–101] backbone dynamics upon binding dU1[SLII] or dU2[SLIV].** Fast timescale (ps-ns) backbone dynamics of dSNF[1–101] are quite similar in the absence or presence of RNA (dU1[SLII] or dU2[SLIV]; Supplementary Fig. 1). C-terminal regions of the protein

are an exception; in the free protein, residues 89–101 (α3 and C-terminus) are mostly disordered on this timescale, as shown by hetNOE experiments. When the hairpins are bound, residues in α3 become less dynamic (Supplementary Fig. 1). Those residues do not

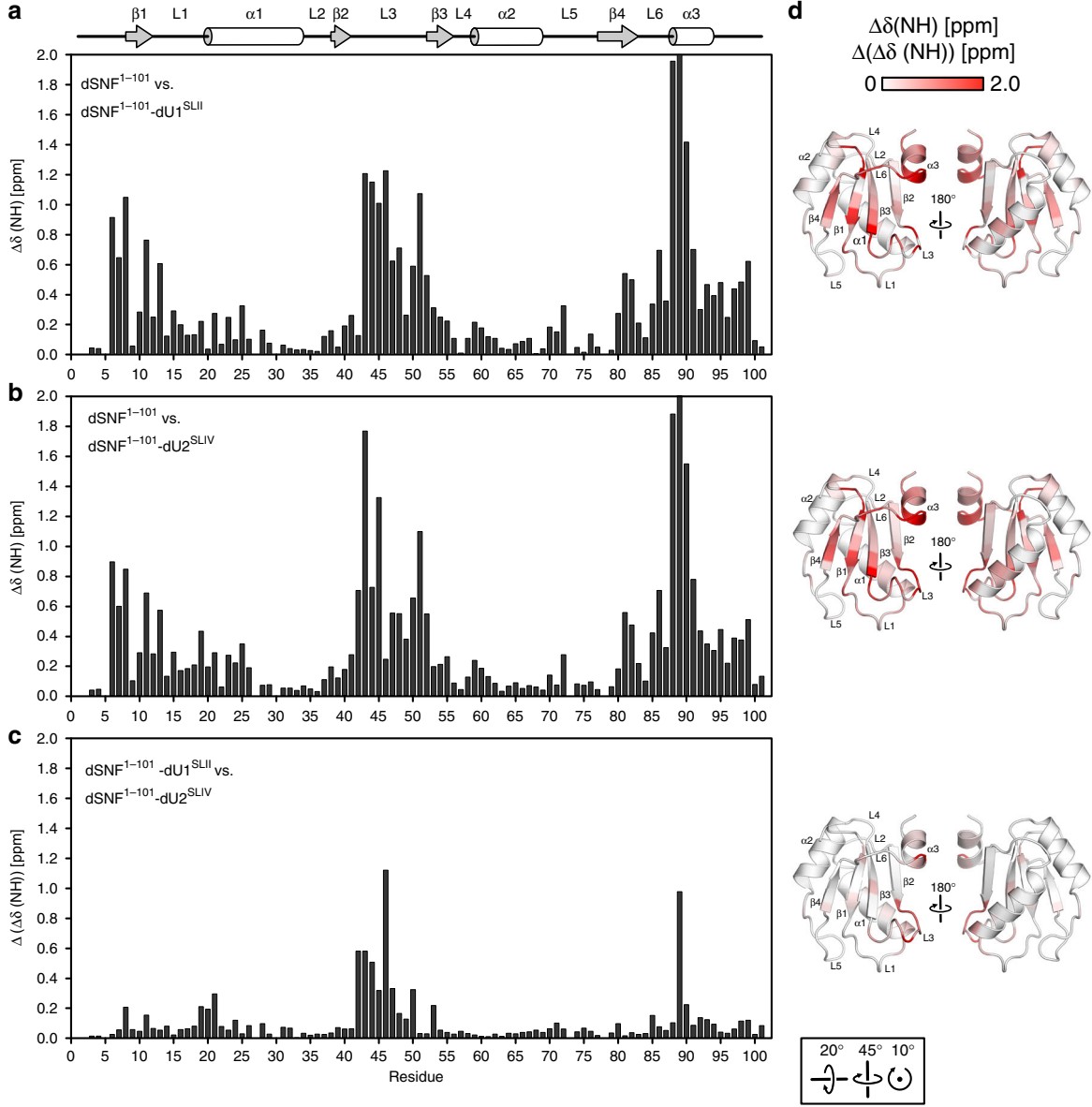

**Fig. 4** dSNF$^{1-101}$-RNA NMR chemical shift perturbations. **a** Backbone amide chemical shift differences between dSNF$^{1-101}$ and dSNF$^{1-101}$–dU1$^{SLII}$ complex. $\Delta\delta = [(\Delta\delta HN)^2 + ((\Delta\delta N)^2*0.154)]^{1/2}$. dSNF$^{1-101}$ at 300 μM, dU1$^{SLII}$ at 700 μM. **b** dSNF$^{1-101}$ backbone amide chemical shift perturbations ($\Delta\delta$) upon dU2$^{SLIV}$ binding. dSNF$^{1-101}$ at 300 μM, dU2$^{SLIV}$ at 1.1 mM. **c** Difference plot of chemical shift perturbations ($\Delta(\Delta\delta) = [\Delta\delta(dSNF^{1-101}-dU2^{SLIV}) - \Delta\delta (dSNF^{1-101}-dU1^{SLII})]$). Positions of secondary structure elements are shown on the top. 50 mM KCl, 20 mM sodium cacodylate, pH 6.5, in 90% $^{1}H_2O$, 10% $^{2}H_2O$; 23 °C; 700 MHz. **d** Orthogonal views of $\Delta\delta$ (top and middle) or $\Delta(\Delta\delta)$ values (bottom) mapped onto the structure of dSNF$^{1-96}$. A scale bar is shown on the top. The orientation of the left panels relative to Fig. 1b is shown at the bottom

make contact with the RNAs, but residues Tyr83-Ser84-Lys85-Ser86-Asp87-Ser88-Asp89 are critical for interactions with nucleotides at the top of the loop, using their backbone amides and carbonyl oxygens to engage in hydrogen bonds to nucleobases and riboses, as also previously described in the hU1A$^{RRM1}$–hU1$^{SLII}$ crystal structure[13]. We suspect that these contacts restrict the motions of α3, anchoring it to the body of the complex.

In contrast, the intermediate timescale (μs-ms) dynamics of dSNF$^{1-101}$ bound to dU1$^{SLII}$ or dU2$^{SLIV}$ are strikingly different compared to isolated dSNF$^{1-101}$ (Fig. 2a–d). In the complexes, backbone amides on the RNA-binding surface of dSNF$^{1-101}$ are no longer dynamic on the μs-ms timescale, and only in α1 and L5 do they retain μs-ms motions (Fig. 2b–d). These findings suggest that dSNF$^{1-101}$ limits the entropic cost of complex formation by retaining high-frequency molecular motions throughout and, in

addition, low-frequency motions in elements not directly involved in RNA binding. Notably, α1 and L5 are the binding sites for dU2A′, suggesting that they need to retain flexibility to adapt to that binding surface (see below).

**dU2A′ binds flexible elements in dSNF$^{RRM1}$–RNA complexes.** We next determined a crystal structure of a dU2A′–dSNF$^{1-96}$ complex at 1.42 Å resolution (Supplementary Table 1), which resembles the arrangement of the two proteins in the hU2A ′–hU2B″$^{RRM1}$–hU2$^{SLIV}$ complex[14]. In the dU2A′–dSNF$^{1-96}$ complex, dSNF$^{1-96}$ helix α1 rests on a shallow, concave surface formed by the parallel β strands of the dU2A′ LRR motif and is held laterally by protruding loops that act like pincers (Fig. 5a), with electrostatic contacts in the periphery and hydrophobic interactions in the center of the interface. L5 of dSNF$^{1-96}$

additionally comes to lie on the C-terminal pincer loop of dU2A′ (Fig. 5a). Thus, dU2A′ directly binds dSNF$^{1-96}$ elements α1 and L5, which remained flexible when dSNF$^{1-101}$ alone binds dU1$^{SLII}$ or dU2$^{SLIV}$ (see above). Superposition of all copies of isolated dSNF$^{1-96}$ on the two crystallographically independent and virtually identical dU2A′–dSNF$^{1-96}$ complexes revealed that, with the exception of the α3 region, the overall conformation of

dSNF$^{1-96}$ does not change upon dU2A′ binding (Fig. 5a). We presently do not know if conformational changes in dU2A′ occur upon complex formation.

We analyzed motions of dSNF$^{1-96}$ alone and in complex with dU2A′ by NMR. When dU2A′ is bound to dSNF$^{1-96}$, the overall pattern of $\Delta R_{2,eff}$ terms throughout the body of dSNF$^{1-96}$ is substantially altered (Fig. 5b–d). Only α3 retains its $\Delta R_{2,eff}$

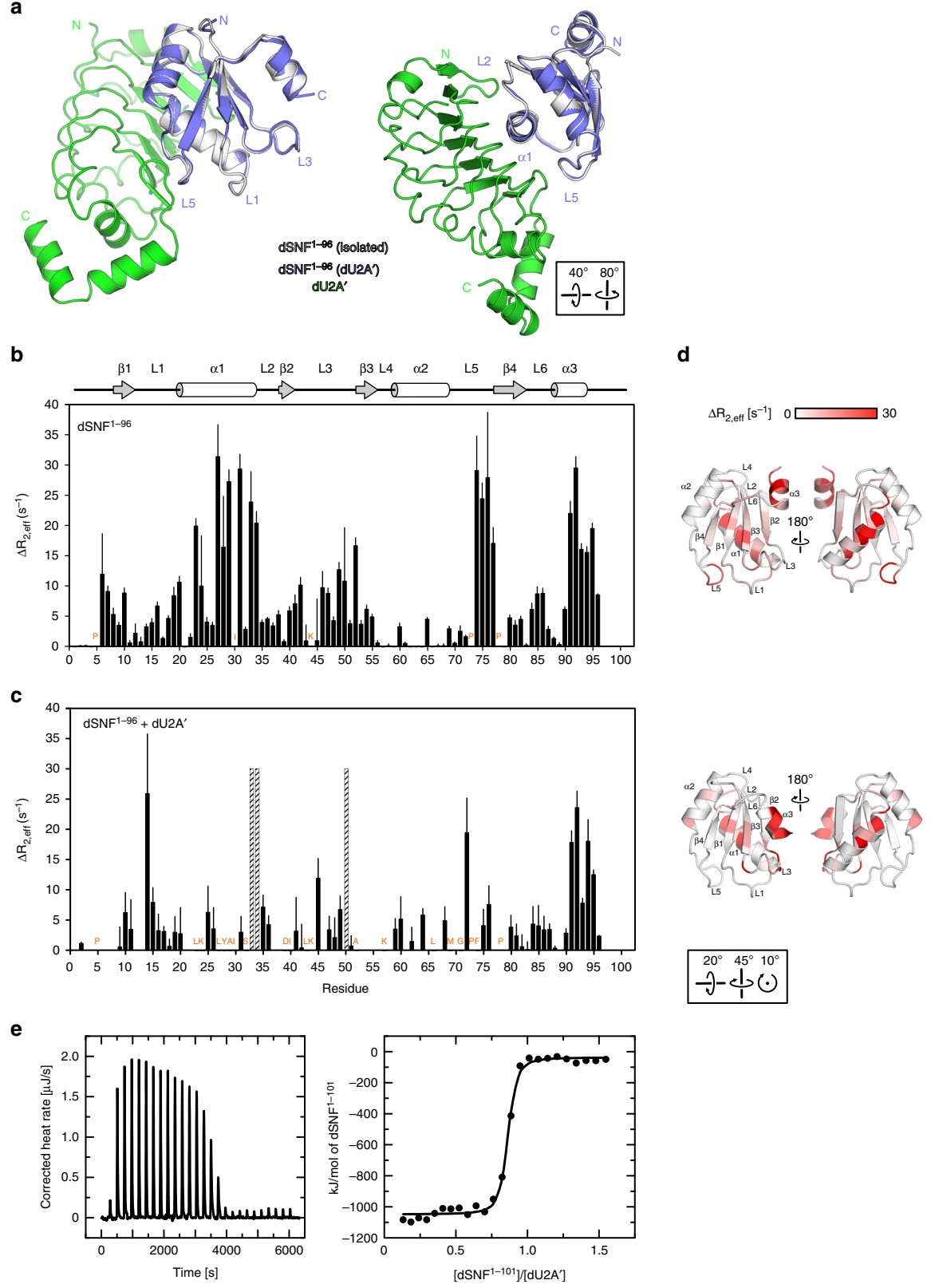

contributions, indicating motions on the µs-ms timescale; many backbone amide resonances within α1 and L5 (labeled orange in Fig. 5c) are absent or too low in intensity to analyze, while amides from L3 do not show evidence of dynamics on this timescale. These samples were not deuterated, so exchange of dSNF[1-96] amide protons with proximal protons from dU2A′ is likely to be responsible for their disappearance in the $\Delta R_{2,eff}$ experiments. These data support the co-crystal structure, where dSNF[1-96] α1 is tightly packed against the surface of dU2A′, and dSNF[1-96] L5 is in contact with the LRR surface. In contrast, backbone amides from L3 and its flanking β2 and β3 do not undergo exchange on this timescale in the binary complex. Consistent with these observations, isothermal titration calorimetry (ITC) experiments measured dU2A′-dSNF[1-101] binding thermodynamics $\Delta H=$ −1008 kJ/mol, $T\Delta S = $ −975 kJ/mol-K ($\Delta S$=−3.3 kJ/mol-K), $K_d$ = 4.2 nM (22 °C; 100 mM arginine, 50 mM KCl, 10 mM sodium cacodylate pH 7; Fig. 5e). The large favorable (negative) enthalpy is offset by the large negative $T\Delta S$ term, which could reflect a loss of conformational flexibility in dSNF[RRM1] and/or dU2A′.

**Structure of a dU2A′–dSNF[1-96]–dU2[SLIV] complex**. As in the dSNF[1-96]–dU1[SLII] complex, the overall structure of dSNF[1-96] is not significantly altered in a crystal structure we determined of a ternary dU2A′–dSNF[1-96]–dU2[SLIV] RNP (Supplementary Table 1; Fig. 6a, b). Significant differences in the shear of the loop-closing U6:G17 wobble pair in U2[SLIV] compared to the C6:G17 Watson-Crick pair in dU1[SLII] lead to a different orientation of the two stem regions on dSNF[1-96] (Fig. 6a). The different positioning of the RNA hairpins is supported by their different interactions with Lys17 and Lys19 (L1; Fig. 6c). Lys17 and Lys19 do not face the backbone of the 5′ branch of dU2[SLIV] (Fig. 6c, top panel) as they do in complex with dU1[SLII] (Fig. 6c, bottom panel). Instead, Lys17 engages in contacts to the major groove side of U6:G17, as well as to G5 of the adjacent base pair of dU2[SLIV]; note that an equivalent interaction of Lys17 to the C6:G17 pair in dU1[SLII] is not equally possible. To enable the Lys17–U6:G17 interaction, the dU2[SLIV] stem must be directed away from Lys19, which lacks dU2[SLIV] contacts (Fig. 6c, top panel). Concomitantly, Arg49 (L3) is displaced from its position next to A7 in dU1[SLII] and instead contacts the phosphate of G17 in dU2[SLIV]. G17 is thereby pulled below the first loop nucleotide (A7; Fig. 6c, top panel), preventing dSNF[1-96] Leu46 from stacking on the loop-closing base pair as in the dU1[SLII] complex (Fig. 6c, bottom panel). No other dSNF[RRM1] contacts to the stem of dU2[SLIV] are seen in the structure.

Notably, a U6C substitution introducing a C:G loop-closing base pair in U2[SLIV] led to threefold tighter binding to dSNF[RRM1] (Fig. 6b), indicating that the interaction network ensuing around a stem oriented as in dU1[SLII] is more stable and thus providing one reason why dU2[SLIV] binding additionally requires U2A′. Furthermore, both types of Lys17-based interactions add similarly to the stability of the respective RNP, as a dSNF[1-101] Lys17Ala variant exhibited a tenfold weaker affinity for both U1[SLII] and U2[SLIV] (Fig. 6b). Thus, due to the different positioning of the

stem, dU2[SLIV] experiences essentially a net loss of the Lys19-based backbone interactions compared to dU1[SLII].

Five loop residues of dU2[SLIV], U8-U9-G10-C11-A12, are sequence-identical to dU1[SLII] and bind dSNF[1-96] in almost the same fashion in the two RNAs. The only difference is that the 6-amino group of A12 is contacted by Ser84 in the dSNF[1-96]–dU1[SLII] complex and by Ser86 in the dU2A′-dSNF[1-96]–dU2[SLIV] complex. The importance of this region is illustrated by a G10A mutation in dU2[SLIV], which led to a 1000-fold reduction in dSNF[1-101] affinity (Fig. 6b).

G13 of dU2[SLIV] adopts a *syn* conformation to occupy an equivalent position as C13 of dU1[SLII], sandwiched by A12 and Asp89 (α3) of dSNF[1-96] and hydrogen bonding to the backbone amide of Asp89 (Fig. 6d). Mutational analysis showed that C13 in dU1[SLII] and G13 in dU2[SLIV] are critical determinants of RNA affinity and specificity. A dU2[SLIV] variant bearing a G13A replacement lacks the ability to support the *syn* conformation by an interaction with its own phosphate and might additionally lose a contact to the Asp89 backbone, resulting in 200-fold reduced affinity to dSNF[1-101] (Fig. 6b). Conversely, a dU1[SLII]-like G13C substitution in dU2[SLIV] led to 16-fold higher affinity for dSNF[1-101] (Fig. 6b), presumably due to the ability of cytidine at this position to additionally contact Asp87 (see above).

Significant differences compared to dU1[SLII] are again seen in the following 3′ sequence of the dU2[SLIV] loop. U14 adopts the same position in both copies of the ternary complex, resembling the minor conformation of the equivalent C14 of the dSNF[1-96]–dU1[SLII] complex, with the base occupying a binding pocket on dSNF[1-96] (Fig. 6d, top and bottom panels). U14 of dU2[SLIV] can engage in more complementary interactions than a cytidine at the equivalent position, with its Watson-Crick hydrogen bonding potential fully saturated by contacts to Lys20, Lys24 (α1) and the backbone carbonyl of Ala42 (L3; Fig. 6d, top panel). Thus, there are more extensive contacts of U14 in dU2[SLIV] to dSNF[1-96] helix α1 compared to C14 in dU1[SLII]. Consistently, a U14C substitution reduced dSNF[1-101] affinity twofold (Fig. 6b). In addition, U14 is held in its position on dSNF[1-96] by Arg20 from the N-terminal pincer loop of dU2A′, acting like a lid (Fig. 6d, top panel).

The additional loop nucleotide A14a in U2[SLIV] forms an extended base stack with the following residues, C15 and C16 (Fig. 6d, top panel). Whereas the equivalent U15 and C16 of dU1[SLII] do not contact dSNF[1-96], the additional nucleotide in dU2[SLIV] allows this RNA region to approach dSNF[1-96] more closely, so that Lys20 (α1) can directly contact the phosphates of C15 and C16 (Fig. 6d, top panel). Stabilization of dSNF[1-96] by dU2A′ (see above) might reinforce these interactions.

**Comparison to the hU2A′–hU2B″[RRM1]–hU2[SLIV] complex**. Globally, the hU2A′–hU2B″[RRM1]–hU2[SLIV] and the dU2A′–dSNF[1-96]–dU2[SLIV] ternary complex structures[14] closely resemble each other. The protein moieties of the *Drosophila* and human complexes superimpose well (rmsd of 0.74 Å for 252 common Cα positions across both proteins; Fig. 7a). All loop nucleotides (A7-

**Fig. 5** Structure and dynamics of the dU2A′–dSNF[1-96] complex. **a** Crystal structure of the dU2A′–dSNF[1-96] complex (dU2A′ - green; dSNF[1-96] - steel blue) in two orientations, superimposed on isolated dSNF[1-96] (gray). The orientation of dSNF[1-96] in the left panel is identical to the view in Fig. 1b, the orientation of the right view is indicated by the boxed rotation symbols. **b** dSNF[1-96] [15]N/[1]H backbone amide dynamics measured with $\Delta R_{2,eff}$ NMR CPMG experiments. dSNF[1-96] is 300 µM in 50 mM KCl, 20 mM sodium cacodylate, pH 6.5, in 90% [1]H$_2$O, 10% [2]H$_2$O; 23 °C; 700 MHz. **c** [15]N-dSNF[1-96] + dU2A′ (1:1) (220 µM/ 220 µM) in 200 mM KCl, 20 mM sodium cacodylate pH 6.5; 23 °C; 700 MHz. Amino acid residues indicated in orange have no detectable signal due to conformational exchange or exchange with proximal protons, hatched bars are line-broadened and not quantifiable. Errors in (**b**) and (**c**) were determined from the propagation of base plane rms noise. **d** Orthogonal views of $\Delta R_{2,eff}$ values mapped onto the structure of dSNF[1-96] in isolation (top) and bound to dU2A′ (bottom). A scale bar is shown on the top. The orientation of the left panels relative to Fig. 1b is shown at the bottom. **e** Representative thermogram and binding isotherm of an ITC experiment assessing binding thermodynamics of the dU2A′-dSNF[1-101] interaction

C16) of hU2$^{SLIV}$ are identical to dU2$^{SLIV}$ and the central and 3′ portions of the loops likewise superimpose well in the two structures. Like dSNF$^{1-96}$, hU2B″$^{RRM1}$ L3 harbors Leu46 and Thr48 (Leu43 and Thr45 in dSNF$^{1-96}$, respectively), which do not permit an extensive water-mediated hydrogen bonding network between the RRM and the RNA, as fostered by the two equivalent

serines (Ser46, Ser48) in the hU2$^{SLII}$–hU1A$^{RRM1}$ structure[13] (Fig. 7b). As a consequence, U14 of the respective U2$^{SLIV}$ is accommodated by dSNF$^{1-96}$ and hU2B″$^{RRM1}$ in the same fashion (Fig. 7b).

A more in-depth comparison revealed subtle differences in U2$^{SLIV}$ binding to dSNF$^{1-96}$ and hU2B″$^{RRM1}$, which arise from

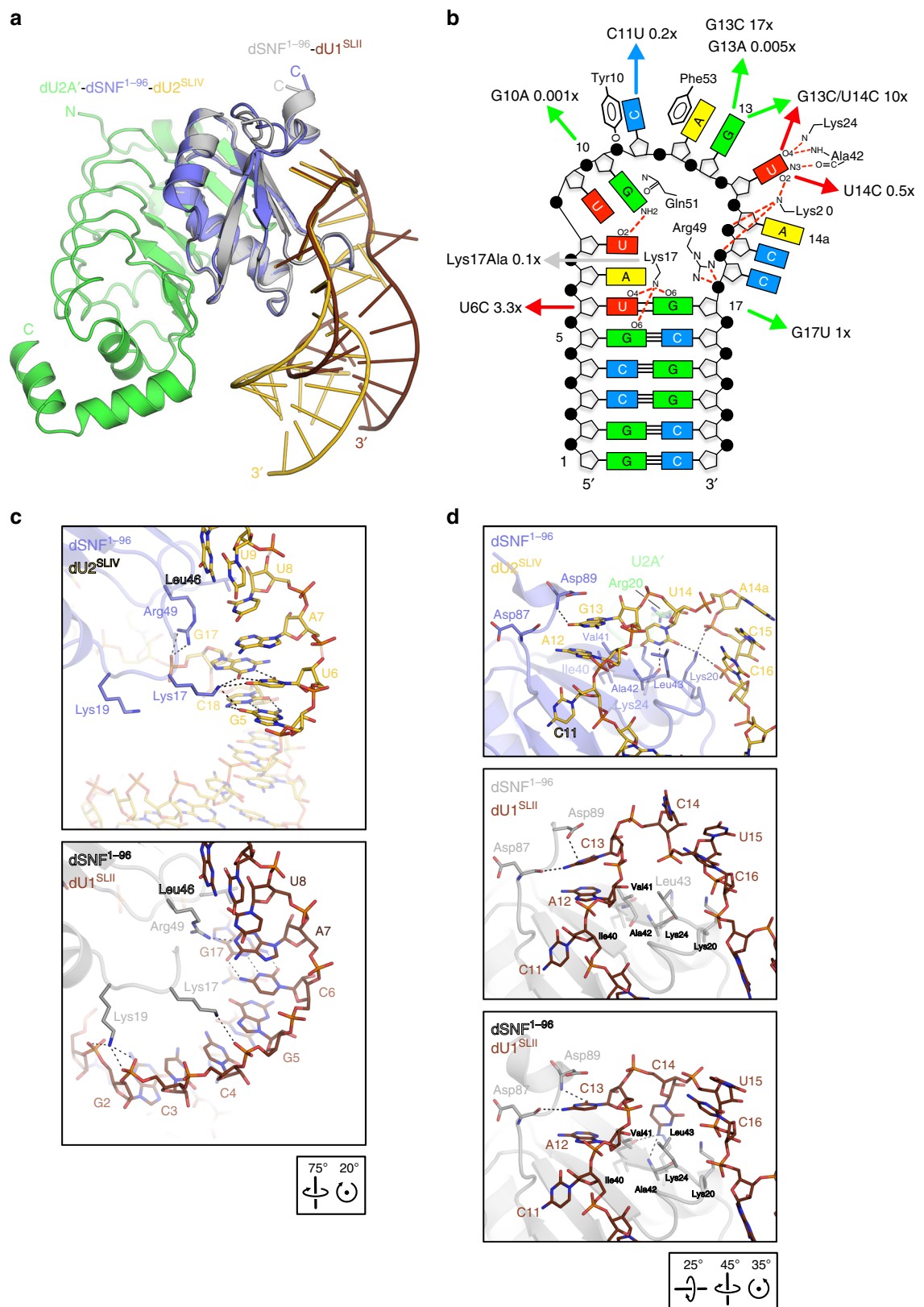

an interplay of the different loop closing base pairs (U6:G17 in dU2$^{SLIV}$; U6:U17 in hU2$^{SLIV}$) and key residue variations between the proteins (Supplementary Fig. 2). On the major groove side, the loop closing base pairs are recognized directly by Lys17/Lys20 (L1) in an equivalent fashion (Fig. 7c). The side chain of hU2B″$^{RRM1}$ Met49 (L3) is longer than that of the equivalent dSNF$^{1-96}$ Leu46 and abuts the U17 ribose of the loop-closing base pair (Fig. 7d). As a consequence, hU2B″$^{RRM1}$ Met49 seems to push U17 of hU1$^{SLIV}$ underneath A7. A7, in turn, is pulled towards U17 by its N6 amino group hydrogen-bonding to Asp19 (L1) of dSNF$^{1-96}$, and so A7 stacks efficiently on the center of the U6:U17 loop-closing base pair (Fig. 7d, right panel). Moreover, Asp19, Arg52 (L3), and G10 interact with the Watson–Crick face of U8, which positions this nucleobase above A7, thereby extending the base stack. In the *Drosophila* system, the ideal stacking position for A7 on the larger U6:G17 loop-closing base pair is farther remote from the dSNF$^{1-96}$ surface (Fig. 7d, left panel). As a consequence, A7 does not directly interact with Glu16 (L1; the equivalent of Asp19 in hU2B$^{RRM1}$). Instead, Glu16 is positioned between U8 and G10, allowing U8 to be positioned more remote from the dSNF$^{1-96}$ surface and in ideal stacking position on A7. As a consequence, U8 loses its interaction with Arg49 (L3; the equivalent of Arg52 in hU2B″$^{RRM1}$).

Due to the slightly different geometry of the loop-closing base pairs, the RNA stems approach the C-terminal end of U2A′ marginally differently. In the *Drosophila* system, both Gln152 and Lys153 contact the RNA backbone between residues G2 and C4 (Fig. 7e, left panel), while in the human system, hU2A′ Lys149 contacts the U2 phosphate (Fig. 7e, right panel). The orientations of the U2$^{SLIV}$ RNA stems also leads to a unique hydrogen bond between hU2B″$^{RRM1}$ Lys22 and C2, which is absent in the equivalent dSNF$^{1-96}$ Lys19. Notably, several of the interaction networks revealed here that control specificity and affinity of dSNF$^{RRM1}$, were proposed by Price et al.[14] based on analysis of the hU2B″$^{RRM1}$–hU2A′–hU2$^{SLIV}$ structure.

**Differential energetic contributions to dSNF$^{RRM1}$-RNA binding.** To compare the energetic driving forces for association of the RNAs to the proteins, we used ITC to directly measure the enthalpy and calculate the entropy and dissociation constants ($K_d$; Table 1; Fig. 8a–d). To avoid partial dissociation of the dU2A′–dSNF$^{1-96}$ complex during the titrations, we conducted the experiments at 10 °C. Under the chosen conditions (150 mM KCl, 1 mM MgCl$_2$, 20 mM HEPES-NaOH, pH 7.5, 5% (v/v) glycerol), binding to both RNAs is enthalpically driven, although entropically unfavorable for dU1$^{SLII}$ and entropically favored for dU2$^{SLIV}$. The affinity of dSNF$^{1-96}$ to dU2$^{SLIV}$ increased about sevenfold in the presence of dU2A′, while dU2A′ increased the affinity of dSNF$^{1-96}$ to dU1$^{SLII}$ only twofold. Notably, in both cases increased binding in the presence of dU2A′ was due to a more favorable (U2$^{SLIV}$) or less unfavorable (U1$^{SLII}$) entropic contributions, while the enthalpic terms were reduced. These observations are consistent with reduced conformational flexibility of dSNF$^{1-96}$ in complex with dU2A′ resulting in a reduced entropic cost of RNA binding. Due to

differential entropy–enthalpy compensation effects, the affinity for U2$^{SLIV}$ is increased relative to U1$^{SLII}$ (sevenfold difference in affinity), compared to the situation without dU2A′ (23-fold difference in affinity; Table 1).

We also used ITC to test the contributions of dU2A′ Arg20 (which contacts U14 in the loop of dU2$^{SLIV}$; Fig. 6d, top panel) and of dU2A′ C-terminal residues Gln152 and Lys153 (which contact the ribose backbone of the dU2$^{SLIV}$ stem; Fig. 7e, left panel). We generated two variants of dU2A′, in which Arg20 was replaced by an alanine (dU2A′$^{Arg20Ala}$) or in which Gln152, Lys153, as well as the preceding Arg143, Lys149, and Lys151 were replaced by alanines (dU2A′$^{mutC}$). Complexes of dU2A′$^{Arg20Ala}$-dSNF$^{1-96}$ and dU2A′$^{mutC}$-dSNF$^{1-96}$ bound dU2$^{SLIV}$ with only slightly reduced affinities compared to wild type dU2A′ (Table 1; Fig. 8e, f). While the dU2A′ variants might exhibit larger effects on RNA affinity at higher temperature, the results show that under certain conditions dU2A′ can enhance dU2$^{SLIV}$ binding by dSNF$^{1-96}$ by modulation of dSNF$^{1-96}$ alone and without fostering additional RNA contacts.

## Discussion

RBPs are often modular, with repeated structural motifs, such as RRMs, that provide opportunities to tune RNA binding affinity and specificity[23]. Alternatively, RNA affinities and specificities of RBPs can be modulated by interacting proteins. For instance, an RRM of the U2AF35 protein can shift the structural equilibrium of the two RRMs of the U2AF65 protein towards a more open conformation to facilitate recognition of weak polypyrimidine tracts in human pre-mRNAs[24]. As another example, the cold shock domain of Upstream-of-N-ras (Unr) and two RRMs of the sex lethal (SXL) protein form an intimately intertwined complex, in which the RNA affinities and specificities of the proteins are mutually reprogrammed to specifically recognize an RNA element in the 3′ untranslated region of *msl2* mRNA during dosage compensation in *Drosophila*[25]. Yet another case is afforded by the spliceosomal Snu13 and Prp31 proteins that recognize 5′ stem-loops in the major spliceosomal U4 and the minor spliceosomal U4atac snRNAs[26]. Snu13 positions its αβα sandwich fold to bind kink-turns in the RNAs using the edge of a helix and sheet to contact residues that are identical in U4 and U4atac[27, 28]. In contrast, Prp31 uses a NOP domain to interact with Snu13 and with different residues in the two RNAs[26, 29]. Different local structures in the capping pentaloops of the RNAs allow them to adapt differently to Prp31, resulting in different complex stabilities[26]. Moreover, Snu13 also binds kink-turns in box C/D snoRNAs[30]. However, in the latter case it is aided by different NOP domain proteins, NOP56/58[29, 31]. In all of the above cases, the cooperating proteins modulate their respective conformations and/or both proteins exhibit RNA contacts that are important for the stability of RNPs. Among the hundreds of RBPs[2, 4, 32], there are certain to be many others whose specificities are modulated through protein–protein interactions.

Here, we studied how the affinity of dSNF$^{RRM1}$ for related RNA hairpin loops is modulated by the LRR domain of the

---

**Fig. 6** Structure comparison of ternary and binary complexes. **a** Ribbon diagram of the ternary dU2A′–dSNF$^{1-96}$–dU2$^{SLIV}$ complex (green/steel blue/gold) superimposed on the binary dSNF$^{1-96}$–dU1$^{SLII}$ complex (gray/brown) according to the dSNF$^{1-96}$ subunits. The orientation of dSNF$^{1-96}$ is as in Fig. 1b. **b** Scheme of selected protein-RNA interactions in the dU2A′–dSNF$^{1-96}$–dU2$^{SLIV}$ complex. dSNF$^{1-101}$ and dU2$^{SLIV}$ mutations and resulting relative binary affinities (assessed in the absence of dU2A′) are indicated by arrows and labels. Because binding data had to be acquired at different salt concentrations to measure affinities accurately, the ratios of the dissociation constants of wild type and mutant dSNF$^{1-101}$ or dU2$^{SLIV}$ are used to report changes. nx = $K_d$(wt)/$K_d$(mut). nx < 1 indicates wt affinity is higher; nx > 1 indicates mutant binds tighter. **c** Comparison of dSNF$^{1-96}$ contacts to the stems and regions around the loop-closing base pairs of dU2$^{SLIV}$ (top panel) and of dU1$^{SLII}$ (bottom panel). **d** Comparison of dSNF$^{1-96}$ contacts to the loops of dU2$^{SLIV}$ (top panel) and of dU1$^{SLII}$ (middle and bottom panel, representing two different complexes in the crystal structure with C14 contacting or turned away from dSNF$^{1-96}$, respectively). Orientations relative to Fig. 1b are indicated by boxed rotation symbols

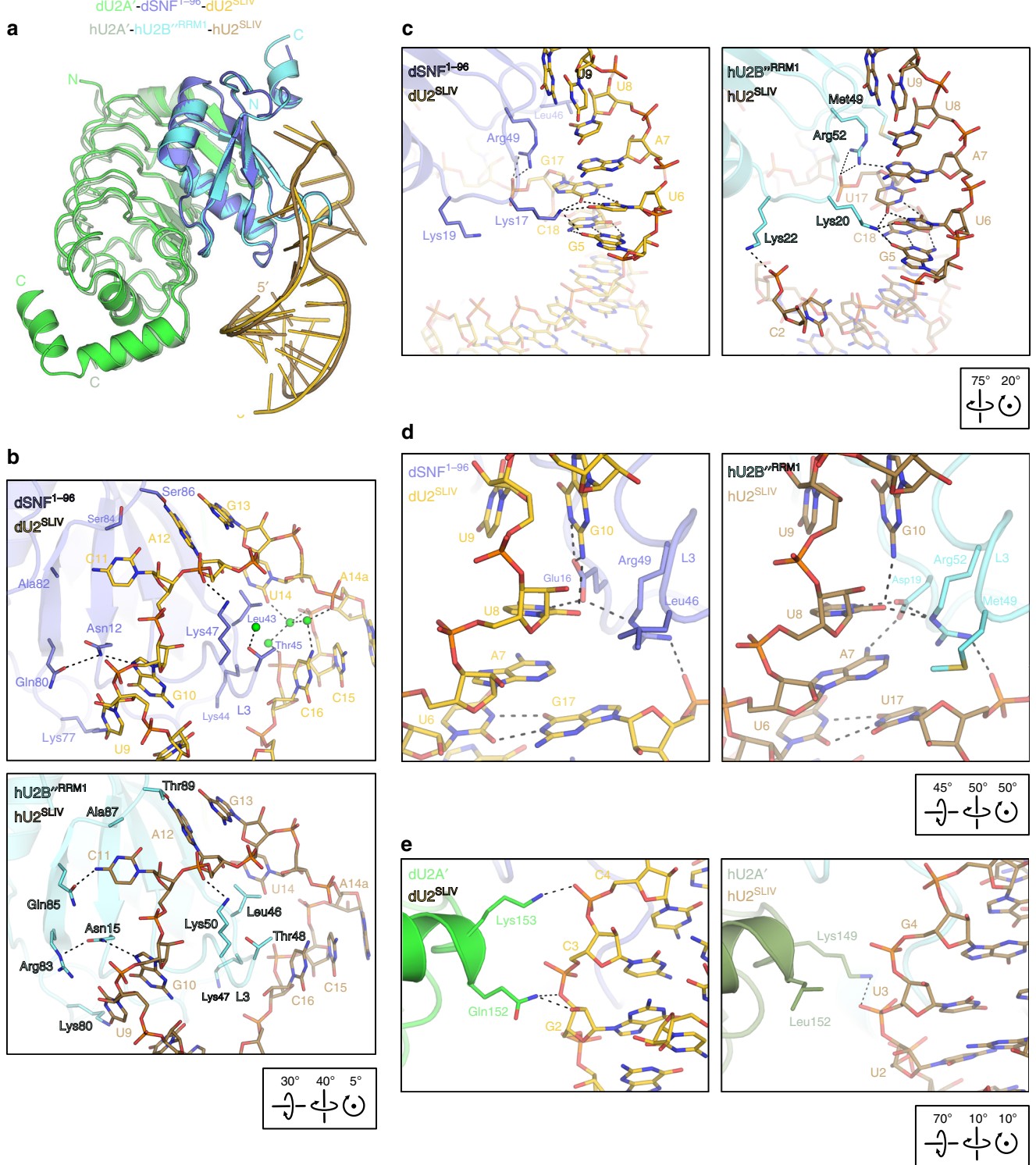

**Fig. 7** Comparison of ternary complexes from *Drosophila* and human. **a** Ribbon diagram of the hU2A′–hU2B″RRM1–hU2SLIV complex (fern green/cyan/light brown; PDB ID 1A9N[14]) superimposed on the dU2A′–dSNF1–96–dU2SLIV complex (green/steel blue/gold) according to the hU2B″RRM1/dSNF1–96 subunits. The orientation of dSNF1–96 is as in Fig. 1b. **b** Contacts to the RNA loops in the *Drosophila* (top) and human (bottom) systems. **c** Interaction networks around the loop-closing base pairs in the *Drosophila* (left) and human (right) systems. **d** Interactions between the loop-closing base pairs and L3 in the *Drosophila* (left) and human (right) systems. **e** Contacts of the respective U2A′ protein to the RNA stems in the *Drosophila* (left) and human (right) systems. Orientations relative to Fig. 1b are indicated by boxed rotation symbols

**Table 1 Thermodynamics of RNA binding**

| | ΔH [kJ/mol] | -TΔS [kJ/mol] | ΔG [kJ/mol] | $K_d$ [nM] | $n$[a] |
|---|---|---|---|---|---|
| $dU1^{SLII}$ | | | | | |
| $dSNF^{1-96}$ | −80.4 | 39.9 | −40.5 | 34.0 | 0.894 |
| $dU2A'$-$dSNF^{1-96}$ | −55.7 | 13.5 | −42.2 | 16.5 | 1.01 |
| $dU2^{SLIV}$ | | | | | |
| $dSNF^{1-96}$ | −30.1 | −3.1 | −33.2 | 782 | 0.781 |
| $dU2A'$-$dSNF^{1-96}$ | −25.4 | −12.2 | −37.6 | 117 | 0.955 |
| $dU2A'^{Arg20Ala}$-$dSNF^{1-96}$ | −20.1 | −16.4 | −36.5 | 187 | 0.939 |
| $dU2A'^{mutC}$-$dSNF^{1-96}$ | −22.6 | −13.7 | −36.3 | 200 | 1.04 |

[a]$n$ stoichiometry, ITC data were evaluated with the PEAQ-ITC analysis software (Malvern Panalytical)

dU2A' protein. We investigated structures and dynamics of $dSNF^{RRM1}$ alone and bound to all of its partners, providing a comprehensive catalogue of its free and bound states. Our results demonstrate that, except for its termini, $dSNF^{RRM1}$ retains the same overall conformation in isolation or when interacting with $dU1^{SLII}$, $dU2A'$ or both $dU2A'$ and $dU2^{SLIV}$ (Cα rmsd 0.39–0.58 Å for dSNF residues 4–83). Moreover, one side of $dSNF^{RRM1}$ binds RNA, while the other side binds dU2A'; while dU2A' fosters contacts to $dU2^{SLIV}$ in the $dU2A'$–$dSNF^{1-96}$–$dU2^{SLIV}$ complex, which are not possible with $dU1^{SLII}$, these interactions do not contribute to higher $dU2^{SLIV}$ affinity under all conditions. Thus, in stark contrast to the cases described above, dU2A' can tune RNA affinity and specificity of $dSNF^{RRM1}$ (i) without inducing a conformational change in $dSNF^{RRM1}$ and (ii) without relying on additional, own RNA interactions.

One reason for the higher affinity of $dSNF^{RRM1}$ for $dU1^{SLII}$ compared to $dU2^{SLIV}$ is a more stable binding of the $dU1^{SLII}$ stem and central loop region by $dSNF^{RRM1}$. Our structures and structure-guided mutational analyses indicate that the stem, the loop-closing C6:G17 base pair and C13 in the loop of $dU1^{SLII}$ interact more intimately with $dSNF^{RRM1}$ than the corresponding stem, U6:G17 wobble pair and loop G13 in $U2^{SLIV}$. Conversely, the 3′ portion of the $dU2^{SLIV}$ loop exhibits higher complementarity to $dSNF^{RRM1}$ than the corresponding region of $dU1^{SLII}$. In particular U14 in $dU2^{SLIV}$ engages in more intimate contacts to $dSNF^{RRM1}$ helix α1 residues (Lys20 and Lys24), as opposed to the equivalent C14 in $dU1^{SLII}$. Furthermore, due to the additional loop nucleotide A14a in $dU2^{SLIV}$, $dSNF^{RRM1}$ α1 residue Lys20 can contact C15 and C16 phosphates in $dU2^{SLIV}$ but not the equivalent U15 and C16 phosphates in $dU1^{SLII}$. While, therefore, the 3′ portion of the $dU2^{SLIV}$ loop can in principle foster more stable interactions with $dSNF^{RRM1}$ than the corresponding region in $U1^{SLII}$, these interactions apparently require presence of dU2A'.

Several structural features of the $dU2A'$–$dSNF^{1-96}$–$dU2^{SLIV}$ complex are fully consistent with NMR chemical shift perturbations seen in the binary $dSNF^{1-101}$–$dU2^{SLIV}$ complex, such as G13 of $dU2^{SLIV}$ contacting Asp89 of $dSNF^{1-96}$, de-stacking of $dSNF^{1-96}$ Leu46 and disruption of the hydrogen bond of its backbone amide to the RNA backbone, hydrogen bonding of Gln51 to RNA and packing of Ala42 against RNA. Therefore, dU2A' apparently does not change the way $dU2^{SLIV}$ is positioned on $dSNF^{1-96}$. Consistent with this notion, under certain conditions dU2A' also enhances interaction of $dSNF^{1-96}$ with $dU1^{SLII}$, albeit to a much smaller extent than the $dSNF^{1-96}$–$dU2^{SLIV}$ interaction.

Based on these observations, we suggest that the complementarity of the 3′ part of the $dU2^{SLIV}$ loop to $dSNF^{RRM1}$

cannot be fully accessed in the binary $dSNF^{RRM1}$–$dU2^{SLIV}$ interaction, most likely because interacting protein and RNA elements remain flexible. As shown by our NMR analyses, $dSNF^{RRM1}$ loses backbone flexibility in complex with dU2A'. In particular, dU2A' rigidifies $dSNF^{RRM1}$ helix α1 and our crystal structure of the ternary complex shows that dU2A' also stabilizes $dU2^{SLIV}$ U14. Formation of a $dU2A'$–$dSNF^{RRM1}$ complex is driven by a large negative enthalpy and opposed by a large negative entropy, consistent with locking out conformational flexibility of both dU2A' and $dSNF^{RRM1}$. Furthermore, subsequent RNA binding is characterized by a less favorable enthalpic contribution than in the binding of isolated $dSNF^{RRM1}$ to RNA, but also by more favorable/less unfavorable interaction entropy. Fully in line with interactions of the 3′ portion of the $dU2^{SLIV}$ loop with $dSNF^{RRM1}$ depending on dU2A'-mediated stabilization, fluorescence measurements after replacement of A14a with 2-amino purine (2AP) showed that addition of $dSNF^{RRM1}$ leads to loss of 2AP stacking, which is recovered upon addition of dU2A'[33]. Our thermodynamic analyses indicate that due to distinct enthalpy–entropy compensation effects, presence of dU2A' enhances the $dSNF^{RRM1}$–$dU2^{SLIV}$ interaction more strongly than the $dSNF^{RRM1}$–$dU1^{SLII}$ interaction.

We presented a mechanism by which the RNA affinity and specificity of an RBP can be modulated via a protein interaction partner without the induction of significant conformational changes and without the second protein engaging in essential RNA contacts itself. In 2014, Gerstberger et al.[2] identified 1542 RBPs, comprising 7.5% of all protein-coding genes in humans. Many RBPs have multiple RNA targets in cells and the same RNA can be bound by several RBPs. Moreover, RBPs typically interact not only with RNAs but also with other proteins. We therefore expect that the principles uncovered here will also play a role in the assembly of many other RNPs. The present example of proteins modulating their RNA affinities and specificities is certain to be one of many in cells, with several other mechanisms already known[34]. The challenge is to recognize when modulation of RBP specificity and affinity occurs, and to understand how these properties are tuned.

## Methods

**Preparation of complexes**. For crystallographic investigations, DNA coding for residues 1–96 of *Drosophila melanogaster* SNF was cloned into pETM-11 using NcoI and XhoI restriction sites to yield a fusion protein with a tobacco etch virus (TEV)-cleavable N-terminal hexa-histidine tag. Production of dmU2A' was based on a triple cysteine variant to prevent non-specific intermolecular disulfide formation, as described before[16]. For co-expression of $dSNF^{1-96}$ and dU2A', a bicistronic expression construct of $dSNF^{1-96}$ and dU2A' open reading frames was assembled by PCR and cloned into pETM-11 using NcoI and XhoI restriction sites. The expression construct comprised DNA coding for a hexa-histidine tagged, TEV-cleavable $dSNF^{1-96}$ followed by untagged dU2A'. Mutations encoding dU2A'

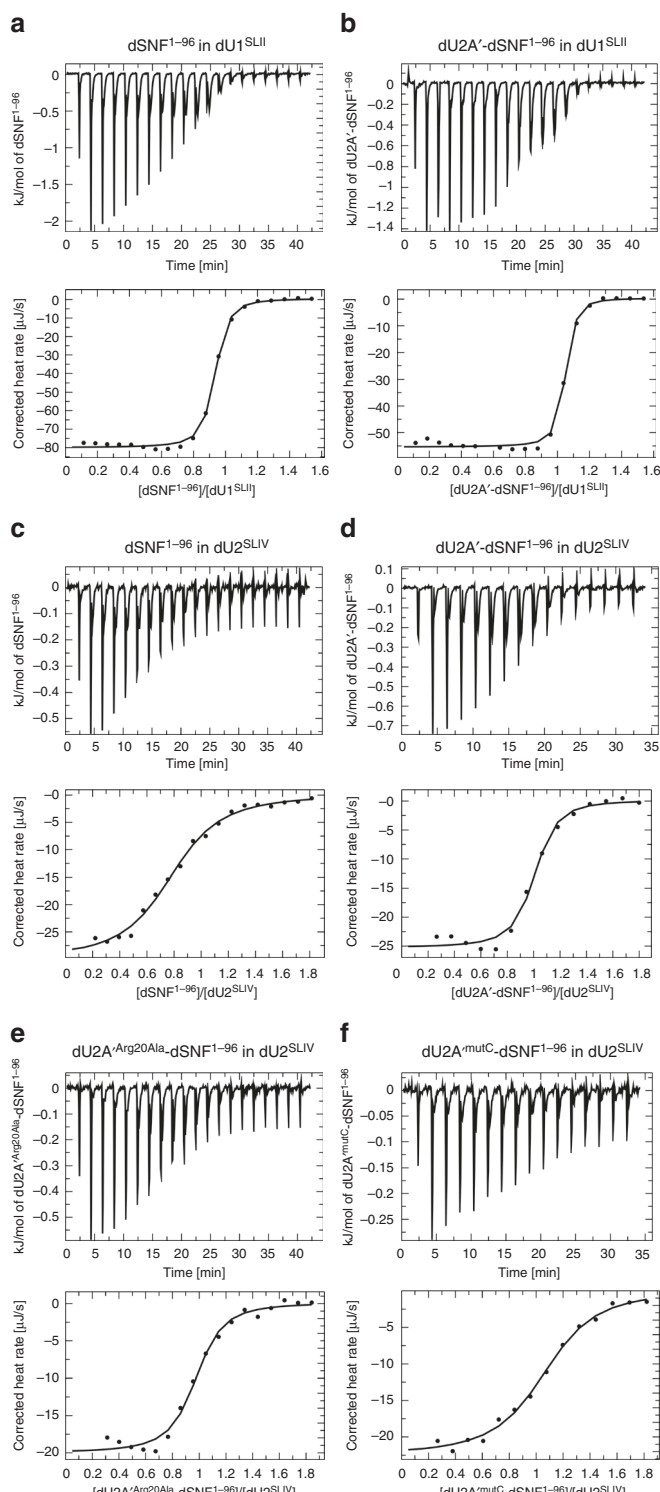

**Fig. 8** ITC analysis of RNA binding by dSNF$^{1-96}$ or dU2A′–dSNF$^{1-96}$. **a–e** Representative thermograms and binding isotherms of ITC experiments assessing the thermodynamics of binding of dSNF$^{1-96}$ (**a**, **c**), dU2A′–dSNF$^{1-96}$ (**b**, **d**), dU2A′$^{Arg20Ala}$–dSNF$^{1-96}$ (**e**), and dU2A′$^{mutC}$–dSNF$^{1-96}$ (**f**) to the indicated RNA hairpins

variants Arg20Ala and Arg143Ala/Lys149Ala/Lys151Ala/Gln152Ala/Lys153Ala (mutC) or SNF variants were introduced by QuikChange Mutagenesis (Agilent). All DNA oligonucleotides used in this study are listed in Supplementary Table 2.

For protein production, *Escherichia coli* BL21 T7 Express cells (NEB) were transformed with pETM-11:dSNF$^{1-96}$ or pETM-11:dSNF$^{1-96}$:dU2A′, grown in

Terrific Broth to an OD$_{600}$ of 0.8 at 37 °C, cooled to 20 °C, induced with 0.5 mM IPTG and incubated at 20 °C overnight. Cells were harvested by centrifugation, resuspended in lysis buffer (400 mM NaCl, 1 mM DTT, 20 mM TRIS–HCl, pH 7.5) and stored at −20 °C. Cells were lysed using a homogenizer and cell debris were removed by centrifugation. All purification steps were carried out at 4 °C.

dSNF$^{1-96}$ was purified by affinity capture on 5 ml Ni-NTA equilibrated in 400 mM NaCl, 8 mM imidazole, 0.5 mM DTT, 20 mM TRIS−HCl, pH 7.5 (buffer A) and elution with a gradient (10 column volumes) to buffer A supplemented with 400 mM imidazole. Eluted fractions were incubated with His-tagged TEV protease during overnight dialysis into buffer A, followed by a recycling step over 1 ml Ni-NTA. The flow-through was diluted to a NaCl-concentration of 100 mM with 0.5 mM DTT, 20 mM TRIS–HCl, pH 7.5. Contaminating nucleic acids were removed by binding the sample to a 20 ml Heparin Sepharose column (GE Healthcare) equilibrated with 100 mM NaCl, 1 mM DTT, 20 mM Tris–HCl, pH 7.5, and eluting the protein with a gradient (5 column volumes) to 1 M NaCl, 1 mM DTT, 20 mM TRIS-HCl, pH 7.5. dSNF$^{1-96}$ was further purified by size exclusion chromatography using a Superdex 75 10/300 gel filtration column (GE Healthcare) in 200 mM NaCl, 1 mM DTT, 20 mM TRIS-HCl, pH 7.5 (gel filtration buffer) to yield pure protein as judged by SDS–PAGE. dSNF$^{1-96}$ was concentrated to 36 mg/ml, flash frozen in liquid nitrogen and stored at −80 °C.

In total 3.1 mg of a synthetic 22-mer dU1$^{SLII}$ RNA (5′-GGCCGC [AUUGCACCUC]GCGGCC-3′; loop in brackets; Agilent Labs) were dissolved in 300 μl Milli-Q water, incubated at 65 °C for 5 min, cooled on ice and supplemented with 150 μl gel filtration buffer. For complex assembly, dSNF$^{1-96}$ was mixed with a twofold molar excess of dU1$^{SLII}$ RNA and incubated for 30 min on ice. The protein-RNA complex was purified by size exclusion chromatography on a Superdex 75 10/300 gel filtration column in gel filtration buffer. dSNF$^{1-96}$–dU1$^{SLII}$ was concentrated to 9 mg/ml, flash frozen in liquid nitrogen and stored at −80 °C.

The dU2A′–dSNF$^{1-96}$ complex (with dU2A′ wild type, dU2A′$^{Arg20Ala}$ or dU2A′$^{mutC}$) was purified, as described for dSNF$^{1-96}$ alone, but all buffers were supplemented with 5% (v/v) glycerol and the minimal salt concentration during all steps of purification was 200 mM NaCl. Purification over heparin yielded two peaks, dU2A′–dSNF$^{1-96}$ and free dSNF$^{1-96}$. dU2A′–dSNF$^{1-96}$ fractions were pooled and further purified by size exclusion chromatography using a Superdex 75 10/300 gel filtration column. Purified dU2A′–dSNF$^{1-96}$ was concentrated to 10 mg/ml, flash frozen in liquid nitrogen and stored at −80 °C.

In total 7.8 mg of chemically synthetized 25-mer dU2$^{SLIV}$ RNA (5′-GCGGCCGU[AUUGCAGUACC]GCGGCC-3′; loop in brackets;) were dissolved in gel filtration buffer to yield a concentration of 0.7 mg/ml, incubated at 95 °C for 5 min and cooled on ice. For complex assembly, dU2A′–dSNF$^{1-96}$ was mixed with a twofold molar excess of dU2$^{SLIV}$ RNA, incubated at 25 °C for 10 min and cooled on ice. The protein–RNA complex was purified by size exclusion chromatography on a Superdex 75 10/300 gel filtration column. dU2A′–dSNF$^{1-96}$–dU2$^{SLIV}$ was concentrated to 13 mg/ml, flash frozen in liquid nitrogen and stored at −80 °C.

For NMR and ITC experiments, dSNF$^{1-101}$ was produced and purified, as described previously[21]. The Lys17Ala exchange was introduced via QuikChange Mutagenesis (Supplementary Table 2). Briefly, protein constructs were isolated from *E. coli* BL21(DE3) cells (Invitrogen) that had been transformed with a plasmid carrying the protein of interest under control of the TAC promoter and ampicillin resistance. Cells were grown in LB medium at 37 °C and induced at OD$_{600}$ = 0.6–0.8 with 1 mM IPTG, then grown for an additional 4 h at 30 °C. Proteins for NMR experiments were grown in minimal media supplemented with either $^{15}$NH$_4$Cl and/or $^{13}$C-glucose. Cells were pelleted and stored at −70 °C until lysis in 100 mM NaCl, 2 mM EDTA, 8.5% (w/v) sucrose, 50 mM sodium acetate, pH 5.3, supplemented with Protease Inhibitor Cocktail (Sigma), PMSF and DNase II. The cell suspension was sonicated then centrifuged. Lysate was passed over an SP-XL Sepharose FPLC column (GE Healthcare), pre-equilibrated in 20 mM sodium cacodylate, pH 7.0, washed with 0 and 100 mM NaCl and eluted over a 100–400 mM NaCl gradient. Fractions containing protein were collected, concentrated and run over a Superdex 75 10/300 gel filtration column in 50 mM KCl, 1 mM EDTA, 20 mM sodium cacodylate, pH 6.5. Desired fractions were concentrated with 10 kDa MWCO Vivaspin concentrators (GE Healthcare). For dSNF$^{1-101}$–RNA complexes, dSNF$^{1-101}$ protein solution was added slowly to purified lyophilized folded RNA to create RNA-bound complexes with either dU1$^{SLII}$ or dU2$^{SLIV}$.

To purify dU2A′-dSNF$^{1-96}$ complex for NMR analyses, dSNF$^{1-96}$ with a hexa-histidine tag at the N-terminal end and dU2A′ were isolated from *Escherichia coli* BL21(DE3) cells that had been transformed with a plasmid carrying the protein of interest under control of the pET promoter and kanamycin resistance. Cells were grown in LB medium (dU2A′) or minimal media supplemented with either $^{15}$NH$_4$Cl and/or $^{13}$C-glucose (dSNF$^{1-96}$) at 37 °C and induced at OD$_{600}$ = 0.8 with 0.5 mM IPTG, then grown an additional 16 h at 25 °C. Cells were pelleted and stored at −70 °C until lysis. Pellets from $^{15}$N or $^{15}$N/$^{13}$C-labeled dSNF$^{1-96}$ and unlabeled dU2A′ were combined in 400 mM NaCl, 5% (v/v) glycerol, 20 mM HEPES-NaOH, pH 7.5 and supplemented with Protease Inhibitor Cocktail (Sigma), PMSF and DNase II. Cell suspension was sonicated then ultra-centrifuged. Lysate was filtered and bound to Ni-NTA beads, washed and eluted with 400 mM imidazole in lysis buffer. The hexa-histidine tag was cleaved with His-TEV protease overnight and removed with His-TEV after running over Ni-NTA beads a second

time. Flow-through containing the unlabeled dU2A'-[15]N or [15]N/[13]C dSNF[1–96] complex was run over a Superdex 75 10/300 gel filtration column in 200 mM KCl, 20 mM sodium cacodylate, pH 6.5. Fractions containing the complex were concentrated with 10 kDa MWCO Vivaspin concentrators.

**RNA preparation.** RNAs for NMR experiments were prepared by in vitro transcription with T7 RNA polymerase (300 U) from double-stranded oligonucleotides (Supplementary Table 2)[35]. Reactions contained 4 mM of each rNTP and 1 mM rGMP (from stock solutions adjusted to pH 7.0), 26 mM MgCl2, 1 mM spermidine, 0.1% (v/v) Triton-X100, 10 mM DTT, 40 mM Tris–HCl, pH 8.4 in 5 ml at 37 °C for 4 h. When solutions became cloudy, 2 μl inorganic pyrophosphatase (Sigma) were added. After addition of EDTA to a final concentration of 30 mM, reactions were vigorously mixed with an equal volume of equilibrated phenol, centrifuged at low speed for 10 min, and the reaction volume was transferred to acid-washed Corex tubes. Sodium acetate was added to 0.3 M final concentration before mixing with 3 × volumes of 100% ethanol. Tubes were stored at −20 °C overnight. Following centrifugation at 12,000 × g and 4 °C for 30 min, the solution was decanted, and pellets were dried and resuspended in 100 μl MilliQ water. After addition of 100 μl formamide, samples were heated to 95 °C for 3 min, quenched on ice and loaded onto 8 M urea/20% polyacrylamide gels in TRIS-Borate-EDTA. Bands were visualized by UV shadowing, cut out and chopped into small chunks, which were immersed in 0.3 M sodium acetate solution. RNA was extracted by soaking overnight at 37 °C with gentle shaking. Solution was removed, transferred to Corex tubes and centrifuged to clear any acrylamide contaminants, then lyophilized. Product was recovered in a minimal volume of MilliQ water, quantified by UV absorbance and frozen at −20 °C. RNA was dialyzed against buffer before use.

For nitrocellulose filter binding experiments, RNAs were synthesized using T7 RNA polymerase from 200 nM DNA oligonucleotides in 25 μL reactions. RNAs were internally labeled with α-[32]P-UTP and α-[32]P-CTP. Transcription products were purified on denaturing polyacrylamide gels, and bands were cut out and soaked overnight in 0.3 M sodium acetate. Solutions were centrifuged to remove residual gel residue, then 10 μg glycogen (Roche) was added and RNA was precipitated by addition of 3 × volumes of 100% ethanol overnight at −20 °C. Recovered RNA was washed with cold 70% ethanol, dried, resuspended in MilliQ water and stored at −20 °C until needed.

**Nitrocellulose filter binding experiments.** Nitrocellulose filter binding experiments were used to determine the affinity of dSNF[1–101] for wild type and mutant dU2[SLIV] RNAs[36]. Nitrocellulose filters (BA-45; Whatman) were prepared by soaking in KCl solutions at concentrations used in binding experiments. Samples for binding experiments contained pM concentrations of RNA, titrated with increasing concentrations of dSNF[1–101]. Binding affinity was calculated using Kaleidagraph to give dissociation constants (Kd) and Gibbs' free energies[36].

**Isothermal titration calorimetry.** ITC experiments measuring binding thermodynamics of the dU2A'–dSNF[1–101] interaction were conducted on a Nano ITC LV (TA Instruments) at 22 °C in 50 mM KCl, 100 mM arginine, 10 mM sodium cacodylate, pH 6.5, using dU2A' as the sample (2.5 μM) and dSNF1-101 as titrant (25 μM). Proteins were dialyzed into ITC buffer. Data were analyzed using Origin software.

ITC experiments measuring thermodynamics of dSNF[1–96] or dU2A'–dSNF[1–96] binding to RNA were conducted on a iTC200 (Malvern Panalytical) at 10 °C in 150 mM KCl, 1 mM MgCl2, 20 mM HEPES-NaOH, pH 7.5, 5% (v/v) glycerol, using the RNAs as samples (8.3–25 μM) and the proteins as titrants (50–150 μM). 100 μM solutions of synthetic RNAs (IBA GmbH; U1[SLII]: 5'-CCAGGACGC [AUUGCACCUC]GCGUCCUGG-3'; U2[SLIV]: 5'-CCAGGACGU [AUUGCAGUACC]GCGUCCUGG-3'; loops in brackets) were incubated at 80 °C for 3 min, snap-cooled on ice and adjusted to ITC buffer. dSNF[1–96] alone or in complex with dU2A' (wild type, dU2A'[Arg20Ala] or dU2A'[mutC]) was dialyzed into ITC buffer. Data were analyzed using the Microcal PEAQ software.

**Crystallographic analyses.** Proteins and complexes were crystallized by sitting drop vapor diffusion (1 μl protein plus 1 μl reservoir for dSNF[1–96] and dU2A'–dSNF[1–96]–dU2[SLIV] or 100 nl protein plus 100 nl reservoir for dSNF[1–96]–dU1[SLII] and dU2A'–dSNF[1–96]) at 4 °C (dSNF[1–96]–dU1[SLII], dU2A'–dSNF[1–96], dU2A'–dSNF[1–96]–dU2[SLIV]) or 20 °C (dSNF[1–96]). dSNF[1–96] crystallized with a reservoir containing 200 mM sodium chloride, 1 M sodium citrate, 100 mM TRIS–HCl, pH 7.5. Crystals were cryo-protected by transfer into reservoir solution supplemented with 20% (v/v) glycerol. The dSNF[1–96]–dU1[SLII] complex crystallized with a reservoir containing 200 mM Li3citrate, 20% (w/v) PEG 3350 and 15 mM NiCl2 as an additive. Crystals were cryo-protected by transfer into reservoir solution supplemented with 10% (v/v) PEG 400. dU2A'–dSNF[1–96] crystallized with a reservoir containing 200 mM Li2SO4, 30% (w/v) PEG 400, 100 mM sodium cacodylate, pH 6.5. The dU2A'–dSNF[1–96]–dU2[SLIV] complex crystallized with a reservoir containing 200 mM Li2SO4, 20% (w/v) PEG 3350 and 15 mM sarcosine as an additive. Crystals were cryo-protected by transfer into reservoir solution supplemented with 10% (w/v) PEG 400. All crystals were incubated in the respective cryo-protecting

solution for 10–30 s and then flash-cooled in liquid nitrogen. Diffraction data were collected at 100 K on beamline 14.2 of the BESSY II storage ring (Berlin, Germany). All diffraction data were processed with XDS[37].

The structure of dSNF[1–96] was solved by molecular replacement using the program PHASER[38] and a homology model generated by HHpred[39] based on the structure of hU1A[RRM1] (PDB ID 1URN[13]). The structure of dSNF[1–96]–dU1[SLII] was solved by molecular replacement with PHASER employing the structure coordinates of hU1A[RRM1] in complex with hU1[SLII] (PDB ID 1URN[13]), in which the hU1A[RRM1] coordinates had been replaced by the dSNF[1–96] structure and the residues of hU1[SLII] had been exchanged for those of dU1[SLII]. The structure of dU2A'–dSNF[1–96] was solved by molecular replacement with PHASER employing the structure coordinates of the hU2A'–hU2B″[RRM1]–hU2[SLIV] complex (PDB ID 1A9N[14]), in which the coordinates of hU2B″[RRM1] had been replaced by the dSNF[1–96] structure, hU2A' had been replaced by a homology model of dU2A' and the RNA had been omitted. The structure of the dU2A'–dSNF[1–96]–dU2[SLIV] complex was solved by molecular replacement with PHASER using the structure coordinates of dU2A'–SNF[1–96], to which an RNA model had been appended based on the structure coordinates of the hU2A'-hU2B″[RRM1]–hU2[SLIV] complex (PDB ID 1A9N[14]). Structural models were completed through alternating rounds of automated refinement using PHENIX.REFINE[40] and manual model building using COOT[41].

**NMR spectroscopy.** NMR data were acquired either on a 700 MHz ([1]H) Varian Inova spectrometer with z-axis pulsed field gradient triple resonance Varian probe, or a 600 MHz ([1]H) Bruker Avance III spectrometer with QCI cryoprobe. dSNF[1–101] and dSNF[1–101]-RNA samples included 50 mM KCl, 2 mM EDTA, 20 mM sodium cacodylate, pH 6.5, 10% (v/v) [2]H2O at 23 °C. dSNF[1–96] and dU2A'–dSNF[1–96] samples included 200 mM KCl, 1 mM EDTA, 20 mM sodium cacodylate, pH 6.5, 10% (v/v) [2]H2O at 23 °C. Temperature was calibrated against a standard methanol NMR sample at 700 MHz, and NMR Thermometer methanol-d4 (Bruker) for 600 MHz cryoprobes. 2,2-dimethyl-2-silapentane-5-sulfonate was used for chemical shift reference. Unbound dSNF[1–96] and dSNF[1–101] were assigned using established 3D triple-resonance backbone NMR experiments (HNCACB and CBCA(CO)NH)[42]. dSNF[1–96] complexed with unlabeled RNA (dU1[SLII] or dU2[SLIV]) or unlabeled dU2A' protein was assigned at 600 MHz using BEST TROSY 3D pulse sequences[22]. In order to collect data in a usable timeframe, 3D experiments were collected with non-uniform sampling techniques and data (FID) reconstructed with Wagner Lab hmsIST additions[43] to NMRPipe. Non-uniform sample schedules were 20–30%. NMR spectra were processed in NMRPipe[44] and all data were analyzed in NMRViewJ.

The weighted change in backbone amide chemical shifts between free dSNF[1–101] and RNA-bound dSNF[1–101] was calculated as:

$$\Delta\delta = \left[ \left(\Delta\delta H^N\right)^2 + \left(\left(\Delta\delta N\right)^2 * n\right) \right]^{1/2} \quad (1)$$

in which $\Delta\delta H^N$ and $\Delta\delta N$ are calculated as $\Delta\delta H^N = (\delta H^N(wt) - \delta H^N(mutant))$ and similarly $\Delta\delta^{15}N = (\delta^{15}N(wt) - \delta^{15}N(mutant))$ in parts per million (ppm) of the magnetic field. $n = 0.154$ is a standard normalization factor that allows the combination of the [1]H and [15]N chemical shift ranges.

Heteronuclear TROSY [15]N-{[1]H} NOE data were acquired in pairs of spectra, one with and one without saturation of the proton resonances. Experiments with proton saturation used a 3 s relaxation delay with 3 s of saturation, while spectra with no saturation used 6 second relaxation delays between scans. Spectra were collected in duplicate, and for each pair, peak intensity ratios (I/I0) were used to calculate the steady-state NOE. Propagation of the base plane noise was used to give the error in the data.

Relaxation dispersion experiments ($\Delta R_{2,\text{eff}}$) were acquired at 700 [1]H MHz and 23 °C using a relaxation-compensated Carr–Purcell–Meiboom–Gill TROSY pulse sequence. Data were collected at two endpoints: at CPMG $\nu_{\text{CPMG}} = 50$ Hz and $\nu_{\text{CPMG}} = 1000$ Hz. In addition, two experiments were acquired: at $T_{\text{relax}} = 40$ ms, and a reference with $\nu_{\text{CPMG}} = 0$ Hz (i.e., no CPMG interval). For all experiments, the relaxation delay was 2.5 seconds. $R_{2,\text{app}}$ values were determined as[45]:

$$R_{2,\text{app}}(\nu_{\text{CPMG}}) = -1/T_{\text{relax}} \cdot \ln[I(\nu_{\text{CPMG}})/I_0] \quad (2)$$

where $T_{\text{relax}} = 40$ ms is the total time for the CPMG refocusing period, $I(\nu_{\text{CPMG}})$ is the peak intensity with CPMG refocusing, and $I_0$ is the peak intensity with no refocusing. To contribute to intrinsic $R_2$ from conformational exchange,

$$\Delta R_{2,\text{eff}} = R_{2,\text{app}}(\nu_{\text{CPMG at 50Hz}}) - R_{2,\text{app}}(\nu_{\text{CPMG at 1000Hz}}) \quad (3)$$

or, for simplification:

$$\Delta R_{2,\text{eff}} = -1/T_{\text{relax}} \cdot \ln[I(\nu_{\text{CPMG at 1000Hz}})/I(\nu_{\text{CPMG at 50Hz}})]. \quad (4)$$

**Data availability**. The coordinates and structure factor data have been deposited in the Protein Data Bank (www.pdb.org) with the accession codes 6F4I (dSNF[1–96]), 6F4J (dU2A′-dSNF[1–96]), 6F4H (dSNF[1–96]–dU1[SLII]) and 6F4G (dU2A′–dSNF[1–96]–dU2[SLIV]). Other data are available from the corresponding authors upon reasonable request.

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

## Acknowledgements

This work was supported in part by grant 1R01 GM096444 from NIH (to K.B.H.) and grant SFB740-3 from the Deutsche Forschungsgemeinschaft (to M.C.W.). RNAs for crystallization were a gift from Agilent labs (Dr. Doug Dellinger). RNAs for NMR were prepared by Michael Rau. We acknowledge access to beamlines BL14.1 and 14.2 of the BESSY II storage ring (Berlin, Germany) via the Joint Berlin MX-Laboratory, funded by the Helmholtz-Zentrum Berlin für Materialien und Energie, the Freie Universität Berlin, the Humboldt-Universität zu Berlin, the Max-Delbrück Centrum für Molekulare Medizin and the Leibniz-Institut für Molekulare Pharmakologie.

## Author contributions

G.W., G.T.D., K.B.H. and M.C.W. designed experiments. G.W., G.T.D., N.H. and K.B.H. performed experiments. G.W., G.T.D., K.B.H. and M.C.W. wrote the manuscript.

## Additional information

**Competing interests:** The authors declare no competing interests.

