## [Peer Review File · Nature Communications]

Reviewers' comments:

Reviewer #1 (Remarks to the Author):

This paper aims to show how the interesting *Drosophila* SNF protein, which recognises U1 and U2 snRNAs differentially depending on the presence or absence of the U2A' protein, achieves its specificity. This is an important question in understanding RRM. Much previous work has shown how in the equivalent human system, two proteins U1A and U2B" substitute for SNF and each respectively binds U1 or U2 snRNA specifically (although U2B" requires U2A' to bind U2 snRNA). These studies, including previous crystal structures, have suggested how U1A and U2B", which have many similarities, have each adapted to specifically bind their partner RNAs. Further understanding of how in *Drosophila* one protein, SNF, can bind both RNAs with modulated specificity, would be an important step in our understanding of how specificity is determined.

Weber et al address this question by solving crystal structures of SNF in isolation and in complex with U1 snRNA, U2A', and U2A' and U2 snRNA. NMR analysis of dynamics is also performed on these various complexes. The amount of work performed is impressive, but the presentation of the results at present makes the conclusions rather unclear. For example, both the title and major points in the abstract (e.g. 'U2 SLIV represents a suboptimal SNF ligand due to its loop-closing base pair') are not clearly reflected in the results and discussion of the paper.

Major comments

The manuscript currently lacks detailed comparisons of the SNF structures to previously solved structures. A key question in the field is how SNF can recognise U1 and U2 snRNAs while other systems requires two different proteins to achieve this. A more in-depth discussion of the differences between the human and *Drosophila* proteins and RNAs could shed some light on this, and figures showing structural superpositions would be helpful. Presently the manuscript makes it unclear which interactions are novel and which were seen before in human. Indeed, many interactions identical to human are presented as though this is their first observation. And notably, a previous NMR structure of *Drosophila* SNF (Hu et al, PLoS one, 2009) is not cited – an unfair omission that should be fixed.

Specifically, the interactions surrounding the SLIV loop-closing U:G wobble pair should be directly compared to those surrounding the U:U pair in the human U2B" structure, since this is proposed to be a major discriminating factor. A related question is that since many interactions observed are identical between human U2B" and SLIV and *Drosophila* SNF and SLIV, does this structure account for the inability of SNF to bind the human SLIV sequence (as reported in Williams and Hall, Biochemistry 2010)?

The authors propose an interesting model for how U2A' enhances the affinity of SNF for U2 SLIV. It is suggested that U2A' pre-immobilises SNF helix $\alpha 1$, thus paying for the entropic cost of SLIV binding. Two lines of evidence are suggested to support this notion: 1) by NMR analysis the ΔR_2 values for most of SNF including helix $\alpha 1$ decrease upon U2A' binding, and 2) by ITC it is shown that SNF has the same enthalpy of binding to U1 and U2 snRNA, but for U2 snRNA the entropic cost is greater.

On point 1), this evidence is not particularly strong. Although there is a trend for decreasing ΔR_2 values in SNF upon U2A' binding, for helix $\alpha 1$ only one residue has a measurable decrease in ΔR_2 (residue 31). The other residues are either unaltered or their signals disappear due to exchange or line-broadening. It should be explained if these missing signals demonstrate inherent rigidity - is the only explanation for such exchange and line-broadening a decrease in flexibility? The type of exchange should also be elaborated on – what sort of conformational changes are proposed to lead to such exchange?

On point 2) the ITC analysis cannot justify the conclusion on line 268 that "U2A' selectively

enhances the affinity of SNF RRM1 for U2SLIV at least in part by entropy compensation" because ITC measurements in the presence of U2A' are not presented. If these data are available they should be included: the prediction from their model is that in the presence of U2A' the enthalpy of binding to SLIV is unaltered but the entropic cost would be lower.

Specific comments

Main text

Line 93: by 'heteronuclear NMR' is 'heteronuclear NOE' meant? If so, reference figure S1 here.

Line 97: reference Figure 5B here so that SNF1-96 and SNF1-101 comparison is clear

The structural description of the interaction between SNF and U1 snRNA is unnecessarily detailed given that the majority of the interactions are identical to those previously observed in the human U1A-RNA cocrystal structure. This also includes the details in figure 3C and D panels I-III. Making this shorter would draw more attention to the new features.

In the discussion of chemical shift perturbation upon U1 or U2 binding, it is suggested that the greater shift perturbation of Leu46 with U2 binding suggests "that SNF differentially recognizes the different configurations of the loop-closing base pairs in the two RNAs" (line 156-157). A simpler explanation would be clearer: e.g. Figure 3D shows the Leu46 amide hydrogen bonds to the phosphate backbone of U1 snRNA G17. This equivalent phosphate is displaced in U2 snRNA so cannot make a hydrogen bond, and this accounts for the difference in shift.

The section "SNF1-101 retains fast dynamics upon binding U1SLII or U2SLIV" (lines 159-177) contributes no new relevant information to the paper. I would suggest removal of this section. If this section remains, then the unscientific word 'designed' should be removed from line 172.

As above for U1 snRNA, the structural description of the interaction between SNF and U2 snRNA is unnecessarily detailed given that the majority of the interactions are identical to those previously observed in the human U2A'-U2B''-RNA cocrystal structure. Making this more concise would draw more attention to the new features.

Line 207: delete 'sequence-specific' – this cannot be true as the same interaction (Lys17 to base carbonyl groups) is seen in the human structure but to a U:U base pair, not a G:C base pair.

Figures

The secondary structure looks misannotated in every figure: helix $\alpha 2$ is 11-13 residues long in previous human and drosophila structures (and 13 residues long in the previous NMR structure of SNF) and is only annotated as 7 residues. The following strand $\beta 4$ is 3-4 residues long in previous structures but as annotated as 7 residues. Figure 1B appears consistent with the previous structures. Therefore the secondary structure diagrams in figures 1,2,4, and 5 should be updated to make $\alpha 2$ longer at the C-terminus and $\beta 4$ shorter at the N-terminus, which shifts the position of L5 towards the C-terminus.

Figure 1: A – It would be useful to highlight the differences between SNF and U1A/U2B'', maybe with colour coding to map regions of SNF that match U1A, or match U2B'', or match neither (e.g. as in Price et al, Nature 1998). B – highlight C terminus, and label $\beta 3a$ as it is not clear where this is within L5.

Figure 2: exchange-broadened or absent residues should be indicated as in Figure 5B/C, both for consistency and to not give the impression that these just have low ΔR_2 values. Error bars should be defined in the legend.

Figure 4: these results may be more clearly shown by colouring the structure by shift perturbation, e.g. red for high perturbation and blue for low.

Figure 5: Error bars should be defined in the legend.

Figure 6: in panel B it would be more helpful for the comparison if Ia and Ib could show the same orientation of SNF. In IIb (and elsewhere) the style makes the depth very difficult to see. The emphasis on G13 and U14 should be removed and normal depth cueing applied. Otherwise U14 appears to be in front of G13, Lys20 in front of everything, and Val41 in front of A12, whereas the opposite is presumably true.

In subpanel IIb, U2A' Arg20 should be labelled

In subpanel III, Gln153 appears to be mislabelled: presumably Gln152 is meant.

An alternative to fixing these problems would be to remove panel B, since most interactions highlighted except Gln152 are identical to human, and replace it with a panel comparing the structure to human. In particular it would be interesting to directly compare the positioning and geometry of the G:U loop-closing pair observed here to the U:U pair seen in the human U2A'-U2B''-U2 snRNA.

Legend to panel C: state whether the measured affinities are in the ternary complex or not. At the moment this is unclear. Also the binding ratios would be clearer in reciprocal: e.g. "U6C 3x", which more intuitively suggests that the mutation increased the affinity.

If experiments were performed with the ternary complex, these would be useful to include. It would be especially interesting to mutate Arg20, Gln152, and Lys153 of U2A'.

Figure 7: the axes should be on the same scale for easier comparison.

References

The reference list is duplicated and reference 16 has the author list duplicated.

Reviewer #2 (Remarks to the Author):

In this manuscript Weber et al. present a detailed structural description of RNA binding of the *Drosophila* SNF protein. The manuscript contains a large amount of data, is technically sound, and a piece of work of high quality.

However, to my opinion, the manuscript would benefit a lot from sharpening the presentation and discussion of the results. I found it, for example, quite confusing that the figures showing the details of SNF - U2A' - RNA complex structures don't seem to match the text. In Figure 6 B, panel III an interaction between U2A' and RNA is shown, but the description is missing in the figure legend as well as a discussion in the text. Are these U2A' - RNA interactions important for the differential interaction of SNF with the U2SLIV and U1SLII RNA? The C-terminal α -helix 3 is most affected by U2A' binding. Are there any contacts between this helix and RNA? Based on figures 4 and 5 there seem to be many changes in α -helix 3 upon both RNA and U2A' binding. Wouldn't these residues in U2A' be the perfect targets for mutational analysis?

Minor points:

- Figure S1. Please add error bars

Response to Reviewer Comments

Reviewer comments are repeated in bold, responses are in regular font, changed text passages are in italics.

Reviewer #1

This paper aims to show how the interesting *Drosophila* SNF protein, which recognizes U1 and U2 snRNAs differentially depending on the presence or absence of the U2A' protein, achieves its specificity. This is an important question in understanding RRM. Much previous work has shown how in the equivalent human system, two proteins U1A and U2B'' substitute for SNF and each respectively binds U1 or U2 snRNA specifically (although U2B'' requires U2A' to bind U2 snRNA). These studies, including previous crystal structures, have suggested how U1A and U2B'', which have many similarities, have each adapted to specifically bind their partner RNAs. Further understanding of how in *Drosophila* one protein, SNF, can bind both RNAs with modulated specificity, would be an important step in our understanding of how specificity is determined.

We thank this reviewer for considering the topic of our work interesting and important.

Weber et al address this question by solving crystal structures of SNF in isolation and in complex with U1 snRNA, U2A', and U2A' and U2 snRNA. NMR analysis of dynamics is also performed on these various complexes. The amount of work performed is impressive, but the presentation of the results at present makes the conclusions rather unclear. For example, both the title and major points in the abstract (e.g. 'U2 SLIV represents a suboptimal SNF ligand due to its loop-closing base pair') are not clearly reflected in the results and discussion of the paper.

We thank this reviewer for the critical evaluation. In the revised manuscript, we have carefully addressed all of the points raised as further detailed below.

Major comments

The manuscript currently lacks detailed comparisons of the SNF structures to previously solved structures. A key question in the field is how SNF can recognize U1

and U2 snRNAs while other systems require two different proteins to achieve this. A more in-depth discussion of the differences between the human and *Drosophila* proteins and RNAs could shed some light on this, and figures showing structural superpositions would be helpful. Presently the manuscript makes it unclear which interactions are novel and which were seen before in human. Indeed, many interactions identical to human are presented as though this is their first observation. And notably, a previous NMR structure of *Drosophila* SNF (Hu et al, PLoS one, 2009) is not cited – an unfair omission that should be fixed.

We agree and apologize for these omissions. In the revised manuscript, we have now carefully analyzed the similarities and differences between the human and *Drosophila* systems, and we included structural comparisons in the revised figures. Structural comparisons to the human system are now included in revised Figure 3, revised Figure 7 and new Figure S2. We now also briefly point out the similarities and emphasize the specific differences to the previously analyzed human structures throughout the revised text. We also apologize for omitting reference to the previously solved NMR structure of *Drosophila* SNF, which is now cited (pg. 4, lines 77-81):

We determined a crystal structure of dSNF^{RRM1} based on a construct containing residues 1-96 of dSNF (dSNF¹⁻⁹⁶) at 1.49 Å resolution (Supplementary Table 1), showing that the protein adopts a classic RRM fold, with Tyr10-Gln51-Phe53 displayed on the surface of its four-stranded anti-parallel β-sheet (Figure 1B), in agreement with the previous NMR structure of dSNF^{RRM1} 19.

Specifically, the interactions surrounding the SLIV loop-closing U:G wobble pair should be directly compared to those surrounding the U:U pair in the human U2B^h structure, since this is proposed to be a major discriminating factor.

We are now including a detailed comparison of the different loop-closing base pairs in the two systems, and provide new figure panels illustrating the similarities and differences (revised Figure 7C,D; new Figure S2). In the revised text, we now mention (pg. 12/13, lines 298-323):

A more in-depth comparison revealed subtle differences in U2^{SLIV} binding to dSNF¹⁻⁹⁶ and hU2B^{hRRM1}, which arise from an interplay of the different loop closing base pairs (U6:G17 in dU2^{SLIV}; U6:U17 in hU2^{SLIV}) and key residue variations between the proteins (Supplementary

Fig. 2). On the major groove side, the loop closing base pairs are recognized directly by Lys17/Lys20 (L1) in an equivalent fashion (Figure 7C). The side chain of hU2B^{RRM1} Met49 (L3) is longer than that of the equivalent dSNF¹⁻⁹⁶ Leu46 and abuts the U17 ribose of the loop-closing base pair (Figure 7D). As a consequence, hU2B^{RRM1} Met49 seems to push U17 of hU1^{SLIV} underneath A7. A7, in turn, is pulled towards U17 by its N6 amino group hydrogen-bonding to Asp19 (L1) of dSNF¹⁻⁹⁶, and so A7 stacks efficiently on the center of the U6:U17 loop-closing base pair (Figure 7D, right panel). Moreover, Asp19, Arg52 (L3) and G10 interact with the Watson-Crick face of U8, which positions this nucleobase above A7, thereby extending the base stack. In the *Drosophila* system, the ideal stacking position for A7 on the larger U6:G17 loop-closing base pair is farther remote from the dSNF¹⁻⁹⁶ surface (Figure 7D, left panel). As a consequence, A7 does not directly interact with Glu16 (L1; the equivalent of Asp19 in hU2B^{RRM1}). Instead, Glu16 is positioned between U8 and G10, allowing U8 to be positioned more remote from the dSNF¹⁻⁹⁶ surface and in ideal stacking position on A7. As a consequence, U8 loses its interaction with Arg49 (L3; the equivalent of Arg52 in hU2B^{RRM1}). Due to the slightly different geometry of the loop-closing base pairs, the RNA stems approach the C-terminal end of U2A' marginally differently. In the *Drosophila* system, both Gln152 and Lys153 contact the RNA backbone between residues G2 and C4 (Figure 7E, left panel), while in the human system, hU2A' Lys149 contacts the U2 phosphate (Figure 7E, right panel). The orientations of the U2^{SLIV} RNA stems also leads to a unique hydrogen bond between hU2B^{RRM1} Lys22 and C2, which is absent in the equivalent dSNF¹⁻⁹⁶ Lys19. Notably, several of the interaction networks revealed here that control specificity and affinity of dSNF^{RRM1}, were proposed by Price et al. ¹⁴ based on analysis of the hU2B^{RRM1}-hU2A'-hU2^{SLIV} structure.

A related question is that since many interactions observed are identical between human U2B^{RRM1} and SLIV and *Drosophila* SNF and SLIV, does this structure account for the inability of SNF to bind the human SLIV sequence (as reported in Williams and Hall, *Biochemistry* 2010)?

This is an experimental error that we never corrected, although our later papers used both human and fly SLIV. We found after publication that our human SLIV construct was prone to dimerization, and in this form, SNF won't bind the loop sequence. We redesigned the construct in subsequent experiments. SNF binds both, with equal affinity, as indeed the mutation G17U shows in Figure 6B.

The authors propose an interesting model for how U2A' enhances the affinity of SNF for U2 SLIV. It is suggested that U2A' pre-immobilises SNF helix $\alpha 1$, thus paying for the entropic cost of SLIV binding. Two lines of evidence are suggested to support this notion: 1) by NMR analysis the ΔR_2 values for most of SNF including helix $\alpha 1$ decrease upon U2A' binding, and 2) by ITC it is shown that SNF has the same enthalpy of binding to U1 and U2 snRNA, but for U2 snRNA the entropic cost is greater.

On point 1), this evidence is not particularly strong. Although there is a trend for decreasing ΔR_2 values in SNF upon U2A' binding, for helix $\alpha 1$ only one residue has a measurable decrease in ΔR_2 (residue 31). The other residues are either unaltered or their signals disappear due to exchange or line-broadening. It should be explained if these missing signals demonstrate inherent rigidity - is the only explanation for such exchange and line-broadening a decrease in flexibility? The type of exchange should also be elaborated on – what sort of conformational changes are proposed to lead to such exchange?

We did not provide sufficient explanation of these data for the reader. In addition to the NMR data, we now also provide an ITC titration measuring binding thermodynamics of dSNF¹⁻⁹⁶ and dU2A'. These results show that this interaction is associated with a large favorable enthalpy that is partly offset by a large negative T Δ S term. We inserted the following explanation for our data (pg. 9/10, lines 212-227):

We analyzed motions of dSNF¹⁻⁹⁶ alone and in complex with dU2A' by NMR. When dU2A' is bound to dSNF¹⁻⁹⁶, the overall pattern of $\Delta R_{2,eff}$ terms throughout the body of dSNF¹⁻⁹⁶ is substantially altered (Figure 5B-D). Only $\alpha 3$ retains its $\Delta R_{2,eff}$ contributions, indicating motions on the μ -ms timescale; many backbone amide resonances within $\alpha 1$ and L5 (labeled orange in Figure 5C) are absent or too low in intensity to analyze, while amides from L3 do not show evidence of dynamics on this timescale. These samples were not deuterated, so exchange of dSNF¹⁻⁹⁶ amide protons with proximal protons from dU2A' is likely to be responsible for their disappearance in the $\Delta R_{2,eff}$ experiments. These data support the co-crystal structure, where dSNF¹⁻⁹⁶ $\alpha 1$ is tightly packed against the surface of dU2A', and dSNF¹⁻⁹⁶ L5 is in contact with the LRR surface. In contrast, backbone amides from L3 and its flanking $\beta 2$ and $\beta 3$ do not undergo exchange on this timescale in the binary complex. Consistent with these observations, isothermal titration calorimetry (ITC) experiments measured dU2A'-dSNF¹⁻¹⁰¹ binding thermodynamics $\Delta H = -1008$ kJ/mol, T Δ S = -975 kJ/mol-K ($\Delta S = -3.3$ kJ/mol-K), $K_d = 4.2$ nM (22° C; 100 mM arginine, 50 mM KCl, 10 mM sodium cacodylate pH 7; Figure 5E). The large favorable (negative) enthalpy is offset by the large negative T Δ S term, which could reflect a loss of conformational flexibility in dSNF^{RRM1} and/or dU2A'.

The precise physical nature of the conformational changes that SNF undergoes cannot be determined from the NMR experiments. They could be fluctuations of the backbone chain in a loop region, a dynamic hydrogen bonding network that connects structural elements, or flopping of a disordered tail.

On point 2) the ITC analysis cannot justify the conclusion on line 268 that “U2A’ selectively enhances the affinity of SNF RRM1 for U2SLIV at least in part by entropy compensation” because ITC measurements in the presence of U2A’ are not presented. If these data are available they should be included: the prediction from their model is that in the presence of U2A’ the enthalpy of binding to SLIV is unaltered but the entropic cost would be lower.

We agree with this assessment of the reviewer. We now report ITC data for dSNF¹⁻⁹⁶ as well as dU2A’-dSNF¹⁻⁹⁶ binding to both hairpins. We had to test various conditions to avoid partial dissociation of the dU2A’-dSNF¹⁻⁹⁶ complex upon titration. Thus, we conducted these experiments at 10° C and in buffer containing a small amount of glycerol. Under these conditions, the absolute thermodynamic values are different from those experiments we had shown in the first version of the manuscript. Also, dU2A’ under these conditions also slightly improves binding of dSNF¹⁻⁹⁶ to dU1^{SLII} (twofold), but still significantly less than to dU2^{SLIV} (sevenfold). More importantly, improvement of RNA binding in the presence of dU2A’ is solely due to a more favorable interaction entropy (actually at some cost of interaction enthalpy). The combined entropy gain/enthalpy loss is much more favorable for dU2^{SLIV} compared to dU1^{SLII}. Overall, dU2A’ reduced the dSNF¹⁻⁹⁶ affinity difference between the two RNAs from about 23-fold to about sevenfold. As mentioned above, we also now report ITC titrations measuring binding thermodynamics of dSNF¹⁻⁹⁶ and dU2A’, showing that this interaction is associated with a large favorable enthalpy that is partly offset by a large negative TΔS term. The latter analysis is consistent with our NMR data and could reflect a loss of conformational flexibility in dSNF^{RRM1} and/or dU2A’ upon their interaction. Thus, our ITC analyses are consistent with our original idea that dU2A’ may partly immobilize dSNF^{RRM1} in a manner favorable for subsequent RNA binding. We report the new ITC data in the new Figure 5E, Figure 8 and Table 1. In addition, we now carefully phrased our conclusions in the revised text (pg. 9/10, lines 219-227):

These data support the co-crystal structure, where dSNF¹⁻⁹⁶ α1 is tightly packed against the surface of dU2A’, and dSNF¹⁻⁹⁶ L5 is in contact with the LRR surface. In contrast, backbone amides from L3 and its flanking β2 and β3 do not undergo exchange on this

timescale in the binary complex. Consistent with these observations, isothermal titration calorimetry (ITC) experiments measured dU2A'-dSNF¹⁻¹⁰¹ binding thermodynamics $\Delta H = -1008$ kJ/mol, $T\Delta S = -975$ kJ/mol-K ($\Delta S = -3.3$ kJ/mol-K), $K_d = 4.2$ nM (22° C; 100 mM arginine, 50 mM KCl, 10 mM sodium cacodylate pH 7; Figure 5E). The large favorable (negative) enthalpy is offset by the large negative $T\Delta S$ term, which could reflect a loss of conformational flexibility in dSNF^{RRM1} and/or dU2A'.

Pg. 14, lines 326-339:

To compare the energetic driving forces for association of the RNAs to the proteins, we used ITC to directly measure the enthalpy and calculate the entropy and dissociation constants (K_d ; Table 1; Figure 8A-D). To avoid partial dissociation of the dU2A'-dSNF¹⁻⁹⁶ complex during the titrations, we conducted the experiments at 10° C. Under the chosen conditions (150 mM KCl, 1 mM MgCl₂, 20 mM HEPES-NaOH, pH 7.5, 5 % (v/v) glycerol), binding to both RNAs is enthalpically driven, although entropically unfavorable for dU1^{SLII} and entropically favored for dU2^{SLIV}. The affinity of dSNF¹⁻⁹⁶ to dU2^{SLIV} increased about sevenfold in the presence of dU2A', while dU2A' increased the affinity of dSNF¹⁻⁹⁶ to dU1^{SLII} only twofold. Notably, in both cases increased binding in the presence of dU2A' was due to a more favorable (U2^{SLIV}) or less unfavorable (U1^{SLII}) entropic contributions, while the enthalpic terms were reduced. These observations are consistent with reduced conformational flexibility of dSNF¹⁻⁹⁶ in complex with dU2A' resulting in a reduced entropic cost of RNA binding. Due to differential entropy-enthalpy compensation effects, the affinity for U2^{SLIV} is increased relative to U1^{SLII} (sevenfold difference in affinity), compared to the situation without dU2A' (23-fold difference in affinity; Table 1).

Pg. 17, lines 407-420:

As shown by our NMR analyses, dSNF^{RRM1} loses backbone flexibility in complex with dU2A'. In particular, dU2A' rigidifies dSNF^{RRM1} helix $\alpha 1$ and our crystal structure of the ternary complex shows that dU2A' also stabilizes dU2^{SLIV} U14. Formation of a dU2A'-dSNF^{RRM1} complex is driven by a large negative enthalpy and opposed by a large negative entropy, consistent with locking out conformational flexibility of both dU2A' and dSNF^{RRM1}. Furthermore, subsequent RNA binding is characterized by a less favorable enthalpic contribution than in the binding of isolated dSNF^{RRM1} to RNA, but also by more favorable/less unfavorable interaction entropy. Fully in line with interactions of the 3' portion of the dU2^{SLIV} loop with dSNF^{RRM1} depending on dU2A'-mediated stabilization, fluorescence measurements after replacement of A14a with 2-amino purine (2AP) showed that addition of dSNF^{RRM1} leads

to loss of 2AP stacking, which is recovered upon addition of dU2A' ³³. Our thermodynamic analyses indicate that due to distinct enthalpy-entropy compensation effects, presence of dU2A' enhances the dSNF^{RRM1}-dU2^{SLIV} interaction more strongly than the dSNF^{RRM1}-dU1^{SLII} interaction.

Specific comments

Main text

Line 93: by 'heteronuclear NMR' is 'heteronuclear NOE' meant? If so, reference figure S1 here.

Corrected and reference to Figure S1 included.

Line 97: reference Figure 5B here so that SNF1-96 and SNF1-101 comparison is clear

Done.

The structural description of the interaction between SNF and U1 snRNA is unnecessarily detailed given that the majority of the interactions are identical to those previously observed in the human U1A-RNA cocrystal structure. This also includes the details in figure 3C and D panels I-III. Making this shorter would draw more attention to the new features.

We have massively shortened the passages describing the overall structure and now essentially only focus on the differences between the human and *Drosophila* structures (pg. 5-7, lines 115-151); Figure panels have been adjusted accordingly:

The center of the loop arches across the dSNF¹⁻⁹⁶ β -sheet, L3 protrudes through the RNA loop, while L1, L6 and α 3 border the outside of the loop (Figure 3B). The overall structure closely resembles the previously analyzed hU1A^{RRM1}-hU1^{SLII} complex ¹³ (Figure 3B).

The similar positioning of the U1^{SLII} hairpins on dSNF^{RRM1}/hU1A^{RRM1} is reflected in similar protein contacts to the 5' branches of the RNA stems (Figure 3C). However, molecular interactions in other parts of the complexes differ in detail. In the human system, the side chains of Ser46 and Ser48 together with backbone amides of Arg47 and Lys50 (L3) maintain an extensive, water-mediated hydrogen bonding network with nucleotides A12-G17 (Figure

3D, bottom panel). In contrast in the *Drosophila* complex, the corresponding $dSNF^{RRM1}$ residues (Leu43, Thr45, Lys44 and Lys47, respectively) do not allow the formation of a similarly extensive water network. While Thr45, Lys44 and Lys47 engage in water-mediated contacts to the phosphate oxygens of C16 and G17, a more expanded network is prevented by Leu43, which replaces Ser46 of $hU1A^{RRM1}$ in $dSNF^{RRM1}$ (Figure 3D, top and middle panels). As a consequence, C14 can adopt two alternative conformations. In one conformation, it is bulged out and does not contact $dSNF^{RRM1}$ (two of the three crystallographically independent complexes in the crystal; Figure 3D, top panel); in the other, its base is flipped towards $dSNF^{RRM1}$ (Figure 3D, middle panel), where it rests on a hydrophobic surface formed by Val41 ($\beta 2$) and Leu43 (L3) and where it engages in hydrogen bonds to the side chain of Lys24 ($\alpha 1$) and the backbone carbonyl group of Ile40 ($\beta 2$; one of the crystallographically independent complexes; Figure 3E). The corresponding U14 in the human system is disordered in two of the observed complexes (PDB ID 1URN)¹³; in the third, it is stabilized above C15 by water-mediated interactions to the phosphate of C13 and a direct contact to the base of C16 (Figure 3D, bottom panel). Lys47 (L3) contacts the phosphate on the 5' side of A12 in the *Drosophila* complex (Figure 3D, top and middle panels), while the equivalent $hU1A^{RRM1}$ Lys50 does not (Figure 3D, bottom panel).

Several residues on the RRM β -strand surface and in the C-terminal extension of the RRM are unique to $dSNF$ and lead to different contacts to the RNA loops in *Drosophila* and human (Figure 3D). While Asn12 (Asn15 in $hU1A^{RRM1}$) hydrogen bonds to N7 of G10 in both organisms, it is stabilized by Gln80 (Arg83 in $hU1A^{RRM1}$) only in *Drosophila*. Gln85 of $hU1A^{RRM1}$ hydrogen bonds to the C11 4-amino group, while the corresponding Ala82 in *Drosophila* does not allow for a similar interaction. Conversely, Ser84 in $dSNF^{RRM1}$ engages in a hydrogen bond to the 6-amino group of A12, while the equivalent Ala88 of $hU1A^{RRM1}$ cannot. Instead, the 6-amino group of A12 and the 4-amino group of C13 are jointly bound by Thr89 in $hU1A^{RRM1}$, while the equivalent Ser86 does not engage in such interactions in *Drosophila*. Despite these differences in detail, our structural data are consistent with similar patterns of recognition by $dSNF^{RRM1}$ and $hU1A^{RRM1}$ for their respective $U1^{SLII}$ ²¹.

In the discussion of chemical shift perturbation upon U1 or U2 binding, it is suggested that the greater shift perturbation of Leu46 with U2 binding suggests “that SNF differentially recognizes the different configurations of the loop-closing base pairs in the two RNAs” (line 156-157). A simpler explanation would be clearer: e.g. Figure 3D shows the Leu46 amide hydrogen bonds to the phosphate backbone of U1 snRNA G17. This equivalent phosphate is displaced in U2 snRNA so cannot make a hydrogen bond, and this accounts for the difference in shift.

We thank the reviewer for pointing this out. We now mention and discuss this point in the revised text (pg. 7/8, lines 168-171):

The most prominent differences are seen for Leu46 (L3), whose backbone amide hydrogen bonds to the phosphate of G17 and whose side chain stacks on the G17 base of the loop-closing base pair in the dSNF¹⁻⁹⁶-dU1^{SLII} complex, and Asp89 (α 3) that caps C13 in the dSNF¹⁻⁹⁶-dU1^{SLII} complex.

Pg. 16, lines 397-401:

Several structural features of the dU2A'-dSNF¹⁻⁹⁶-dU2^{SLIV} complex are fully consistent with NMR chemical shift perturbations seen in the binary dSNF¹⁻¹⁰¹-dU2^{SLIV} complex, such as G13 of dU2^{SLIV} contacting Asp89 of dSNF¹⁻⁹⁶, de-stacking of dSNF¹⁻⁹⁶ Leu46 and disruption of the hydrogen bond of its backbone amide to the RNA backbone, hydrogen bonding of Gln51 to RNA and packing of Ala42 against RNA.

The section “SNF1-101 retains fast dynamics upon binding U1SLII or U2SLIV” (lines 159-177) contributes no new relevant information to the paper. I would suggest removal of this section. If this section remains, then the unscientific word ‘designed’ should be removed from line 172.

We believe that this section is essential for the ultimate model for how dU2A' modulates dSNF RNA specificity that we develop. We therefore have retained the section but have streamlined it and avoided the word "designed" (pg. 8, lines 178-196):

dSNF¹⁻¹⁰¹ backbone dynamics upon binding dU1^{SLII} or dU2^{SLIV}

Fast timescale (ps-ns) backbone dynamics of dSNF¹⁻¹⁰¹ are quite similar in the absence or presence of RNA (dU1^{SLII} or dU2^{SLIV}; Supplementary Fig. 1). C-terminal regions of the protein are an exception; in the free protein, residues 89-101 (α 3 and C-terminus) are mostly disordered on this timescale, as shown by hetNOE experiments. When the hairpins are bound, residues in α 3 become less dynamic (Supplementary Fig. 1). Those residues do not make contact with the RNAs, but residues Tyr83-Ser84-Lys85-Ser86-Asp87-Ser88-Asp89 are critical for interactions with nucleotides at the top of the loop, using their backbone amides and carbonyl oxygens to engage in hydrogen bonds to nucleobases and riboses, as

also previously described in the $hU1A^{RRM1}$ - $hU1^{SLII}$ crystal structure¹³. We suspect that these contacts restrict the motions of $\alpha 3$, anchoring it to the body of the complex.

In contrast, the intermediate timescale (μ s-ms) dynamics of $dSNF^{1-101}$ bound to $dU1^{SLII}$ or $dU2^{SLIV}$ are strikingly different compared to isolated $dSNF^{1-101}$ (Figure 2A-D). In the complexes, backbone amides on the RNA-binding surface of $dSNF^{1-101}$ are no longer dynamic on the μ s-ms timescale, and only in $\alpha 1$ and L5 do they retain μ s-ms motions (Figure 2B-D). These findings suggest that $dSNF^{1-101}$ limits the entropic cost of complex formation by retaining high-frequency molecular motions throughout and, in addition, low-frequency motions in elements not directly involved in RNA binding. Notably, $\alpha 1$ and L5 are the binding sites for $dU2A'$, suggesting that they need to retain flexibility to adapt to that binding surface (see below).

As above for U1 snRNA, the structural description of the interaction between SNF and U2 snRNA is unnecessarily detailed given that the majority of the interactions are identical to those previously observed in the human U2A'-U2B''-RNA cocrystal structure. Making this more concise would draw more attention to the new features.

We agree and have again focused our revised description essentially exclusively on the comparison to the $dSNF^{1-96}$ - $dU1^{SLII}$ structure and in particular on the differences in the human and *Drosophila* systems (pg. 12/13, lines 288-323):

*Globally, the $hU2A'$ - $hU2B''^{RRM1}$ - $hU2^{SLIV}$ and the $dU2A'$ - $dSNF^{1-96}$ - $dU2^{SLIV}$ ternary complex structures¹⁴ closely resemble each other. The protein moieties of the *Drosophila* and human complexes superimpose well (rmsd of 0.74 Å for 252 common Ca positions across both proteins; Figure 7A). All loop nucleotides (A7-C16) of $hU2^{SLIV}$ are identical to $dU2^{SLIV}$ and the central and 3' portions of the loops likewise superimpose well in the two structures. Like $dSNF^{1-96}$, $hU2B''^{RRM1}$ L3 harbors Leu46 and Thr48 (Leu43 and Thr45 in $dSNF^{1-96}$, respectively), which do not permit an extensive water-mediated hydrogen bonding network between the RRM and the RNA, as fostered by the two equivalent serines (Ser46, Ser48) in the $hU2^{SLII}$ - $hU1A^{RRM1}$ structure¹³ (Figure 7B). As a consequence, U14 of the respective $U2^{SLIV}$ is accommodated by $dSNF^{1-96}$ and $hU2B''^{RRM1}$ in the same fashion (Figure 7B).*

A more in-depth comparison revealed subtle differences in $U2^{SLIV}$ binding to $dSNF^{1-96}$ and $hU2B''^{RRM1}$, which arise from an interplay of the different loop closing base pairs (U6:G17 in $dU2^{SLIV}$; U6:U17 in $hU2^{SLIV}$) and key residue variations between the proteins (Supplementary Fig. 2). On the major groove side, the loop closing base pairs are recognized directly by Lys17/Lys20 (L1) in an equivalent fashion (Figure 7C). The side chain of $hU2B''^{RRM1}$ Met49 (L3) is longer than that of the equivalent $dSNF^{1-96}$ Leu46 and abuts the U17 ribose of the

loop-closing base pair (Figure 7D). As a consequence, hU2B^{'RRM1} Met49 seems to push U17 of hU1^{SLIV} underneath A7. A7, in turn, is pulled towards U17 by its N6 amino group hydrogen-bonding to Asp19 (L1) of dSNF¹⁻⁹⁶, and so A7 stacks efficiently on the center of the U6:U17 loop-closing base pair (Figure 7D, right panel). Moreover, Asp19, Arg52 (L3) and G10 interact with the Watson-Crick face of U8, which positions this nucleobase above A7, thereby extending the base stack. In the *Drosophila* system, the ideal stacking position for A7 on the larger U6:G17 loop-closing base pair is farther remote from the dSNF¹⁻⁹⁶ surface (Figure 7D, left panel). As a consequence, A7 does not directly interact with Glu16 (L1; the equivalent of Asp19 in hU2B^{'RRM1}). Instead, Glu16 is positioned between U8 and G10, allowing U8 to be positioned more remote from the dSNF¹⁻⁹⁶ surface and in ideal stacking position on A7. As a consequence, U8 loses its interaction with Arg49 (L3; the equivalent of Arg52 in hU2B^{'RRM1}).

Due to the slightly different geometry of the loop-closing base pairs, the RNA stems approach the C-terminal end of U2A' marginally differently. In the *Drosophila* system, both Gln152 and Lys153 contact the RNA backbone between residues G2 and C4 (Figure 7E, left panel), while in the human system, hU2A' Lys149 contacts the U2 phosphate (Figure 7E, right panel). The orientations of the U2^{SLIV} RNA stems also leads to a unique hydrogen bond between hU2B^{'RRM1} Lys22 and C2, which is absent in the equivalent dSNF¹⁻⁹⁶ Lys19. Notably, several of the interaction networks revealed here that control specificity and affinity of dSNF^{RRM1}, were proposed by Price et al. ¹⁴ based on analysis of the hU2B^{'RRM1}-hU2A'-hU2^{SLIV} structure.

Line 207: delete 'sequence-specific' – this cannot be true as the same interaction (Lys17 to base carbonyl groups) is seen in the human structure but to a U:U base pair, not a G:C base pair.

We agree and have removed the expression.

Figures

The secondary structure looks misannotated in every figure: helix $\alpha 2$ is 11-13 residues long in previous human and *Drosophila* structures (and 13 residues long in the previous NMR structure of SNF) and is only annotated as 7 residues. The following strand $\beta 4$ is 3-4 residues long in previous structures but as annotated as 7 residues. Figure 1B appears consistent with the previous structures. Therefore the secondary structure diagrams in figures 1,2,4, and 5 should be updated to make $\alpha 2$ longer at the

C-terminus and $\beta 4$ shorter at the N-terminus, which shifts the position of L5 towards the C-terminus.

We apologize for these mistakes and inconsistencies. We have now carefully re-annotated the secondary structure elements based on our high-resolution dSNF¹⁻⁹⁶ crystal structure. Indeed, this annotation is in close agreement to previous analyses.

Figure 1: A – It would be useful to highlight the differences between SNF and U1A/U2B”, maybe with color coding to map regions of SNF that match U1A, or match U2B”, or match neither (e.g. as in Price et al, Nature 1998). B – highlight C terminus, and label $\beta 3a$ as it is not clear where this is within L5.

We now specifically included descriptions of the differences in the human and *Drosophila* systems in the revised text (see above). In addition, we now compare the systems side-by-side or overlaid in new panels to the revised Figure 3 (panel D), Figure 7 (panels A-E) and Figure S2.

Figure 2: exchange-broadened or absent residues should be indicated as in Figure 5B/C, both for consistency and to not give the impression that these just have low $\Delta R2$ values. Error bars should be defined in the legend.

We agree and have adjusted the figure and the legend accordingly.

Figure 4: these results may be more clearly shown by coloring the structure by shift perturbation, e.g. red for high perturbation and blue for low.

We have included corresponding presentations in the revised Figure 4.

Figure 5: Error bars should be defined in the legend.

Done.

Figure 6: in panel B it would be more helpful for the comparison if Ia and Ib could show the same orientation of SNF. In IIb (and elsewhere) the style makes the depth very difficult to see. The emphasis on G13 and U14 should be removed and normal depth cueing applied. Otherwise U14 appears to be in front of G13, Lys20 in front of everything, and Val41 in front of A12, whereas the opposite is presumably true.

We have carefully adjusted all Figure panels, taking the comments of the referee into account. In all Figures, we superimposed complexes based on the dSNF¹⁻⁹⁶ (or related hU1A^{RRM1} or hU2B^{RRM1}) subunits on isolated dSNF¹⁻⁹⁶, and now provide overviews with these subunits in the same "standard view". For all panels showing structural details, we now provide rotations relative to this "standard view". We tried to carefully adjust depth cueing. In panels showing structural detail, we chose to show the secondary structure elements in semi-transparent view to have the relevant side chains and backbone atoms stand out. It was not possible to show all situations in the same view but we trust that the present collection of panels clarifies to situations.

In subpanel IIb, U2A' Arg20 should be labeled.

Now done in the top panel of new Figure 6D.

In subpanel III, Gln153 appears to be mislabeled: presumably Gln152 is meant.

Sorry, now corrected in new Figure 7E.

An alternative to fixing these problems would be to remove panel B, since most interactions highlighted except Gln152 are identical to human, and replace it with a panel comparing the structure to human. In particular it would be interesting to directly compare the positioning and geometry of the G:U loop-closing pair observed here to the U:U pair seen in the human U2A'-U2B''-U2 snRNA.

Human and *Drosophila* systems are now compared in new Figure 7, panels A-E. The situations around the loop-closing base pairs are compared in panels C and D, the situations around the C-terminal region of the U2A' proteins are shown in panel E.

Legend to panel C: state whether the measured affinities are in the ternary complex or not. At the moment this is unclear. Also the binding ratios would be clearer in reciprocal: e.g. “U6C 3x”, which more intuitively suggests that the mutation increased the affinity.

These data are now summarized in new Figure 6B. The effects of the mutations were assessed in the binary complex, by titrating dSNF¹⁻¹⁰¹ to U2^{SLIV}, as now stated in the legend.

If experiments were performed with the ternary complex, these would be useful to include. It would be especially interesting to mutate Arg20, Gln152, and Lys153 of U2A'.

We have now produced two dU2A' variants (Arg20Ala and Arg143Ala/Lys149Ala/Lys151Ala/Gln152Ala/Lys153Ala [called "mutC"]). In the second variant, we included changes to additional residues in the vicinity of Gln152 and Lys153, which might provide electrostatic attraction to the RNA. We then assessed the affinities dU2A' wt and variants in complex to dSNF¹⁻⁹⁶ to dU1^{SLII} and dU2^{SLIV} in comparison to dSNF¹⁻⁹⁶ alone by ITC. Results are reported in new Table 1 and new Figure 8. We included a brief description of these results in the revised text (pg. 14, lines 340-349):

We also used ITC to test the contributions of dU2A' Arg20 (which contacts U14 in the loop of dU2^{SLIV}; Figure 6D, top panel) and of dU2A' C-terminal residues Gln152 and Lys153 (which contact the ribose backbone of the dU2^{SLIV} stem; Figure 7E, left panel). We generated two variants of dU2A', in which Arg20 was replaced by an alanine (dU2A'^{Arg20Ala}) or in which Gln152, Lys153 as well as the preceding Arg143, Lys149 and Lys151 were replaced by alanines (dU2A'^{mutC}). Complexes of dU2A'^{Arg20Ala}-dSNF¹⁻⁹⁶ and dU2A'^{mutC}-dSNF¹⁻⁹⁶ bound dU2^{SLIV} with only slightly reduced affinities compared to wild type dU2A' (Table 1; Figure 8E,F). While the dU2A' variants might exhibit larger effects on RNA affinity at higher temperature, the results show that under certain conditions dU2A' can enhance dU2^{SLIV} binding by dSNF¹⁻⁹⁶ by modulation of dSNF¹⁻⁹⁶ alone and without fostering additional RNA contacts.

Figure 7: the axes should be on the same scale for easier comparison.

Data from original Figure 7 have been replaced by new data shown in new Figure 8. We chose to keep different scales as heats released for the different complexes are quite different. Direct comparisons of the binding thermodynamics and K_d 's are provided in new Table 1.

References

The reference list is duplicated and reference 16 has the author list duplicated.

Sorry for the mistakes. Corrected.

Reviewer #2

In this manuscript Weber et al. present a detailed structural description of RNA binding of the Drosophila SNF protein. The manuscript contains a large amount of data, is technically sound, and a piece of work of high quality.

We thank the reviewer for their positive comments on our work.

However, to my opinion, the manuscript would benefit a lot from sharpening the presentation and discussion of the results. I found it, for example, quite confusing that the figures showing the details of SNF - U2A' – RNA complex structures don't seem to match the text. In Figure 6 B, panel III an interaction between U2A' and RNA is shown, but the description is missing in the figure legend as well as a discussion in the text. Are these U2A' – RNA interactions important for the differential interaction of SNF with the U2SLIV and U1SLII RNA? The C-terminal α -helix 3 is most affected by U2A' binding. Are there any contacts between this helix and RNA? Based on figures 4 and 5 there seem to be many changes in α -helix 3 upon both RNA and U2A' binding. Wouldn't these residues in U2A' be the perfect targets for mutational analysis?

We apologize for the inconsistencies in our presentation. We have carefully re-written the manuscript and made sure that all figures are described in the legends and are picked up

upon in the revised text. Only the very beginning of helix $\alpha 3$ of dSNF (Asp89) engages in RNA contacts, as now shown in new Figure 6D and mentioned in the revised text (pg. 11, lines 260-262):

G13 of dU2^{SLIV} adopts a syn conformation to occupy an equivalent position as C13 of dU1^{SLII}, sandwiched by A12 and Asp89 ($\alpha 3$) of dSNF¹⁻⁹⁶ and hydrogen bonding to the backbone amide of Asp89 (Figure 6D).

dSNF¹⁻⁹⁶ helix $\alpha 3$ rests on the β -sheet surface of dSNF¹⁻⁹⁶ when the protein is bound to dU2A'. As binding of dU2A' stabilizes dSNF (which we interpret as a major cause for modulation of subsequent RNA binding), one might initially think that this effect could also contribute to the positioning of dSNF¹⁻⁹⁶ helix $\alpha 3$ in the binary dU2A'-dSNF¹⁻⁹⁶ complex structure. However, we also observe this helix $\alpha 3$ conformation in two copies of dSNF¹⁻⁹⁶ in the crystal structure of isolated dSNF¹⁻⁹⁶ and our NMR data indicate that helix $\alpha 3$ remains mobile in the dU2A'-dSNF¹⁻⁹⁶ complex. Thus, we presently do not attribute a particular relevance to the helix $\alpha 3$ conformation seen in the dU2A'-dSNF¹⁻⁹⁶ complex. To avoid confusion, we slightly modified Figure 5A, now showing superposition with one of the isolated dSNF¹⁻⁹⁶ copies, in which the helix $\alpha 3$ conformation is similar to its position in the dU2A'-dSNF¹⁻⁹⁶ complex.

As suggested, we have now produced two dU2A' variants (Arg20Ala and Arg143Ala/Lys149Ala/Lys151Ala/Gln152Ala/Lys153Ala [called "mutC"]). In the second variant, we included changes to additional residues in the vicinity of Gln152 and Lys153, which might provide electrostatic attraction to the RNA. We then assessed the affinities dU2A' wt and variants in complex to dSNF¹⁻⁹⁶ to dU1^{SLII} and dU2^{SLIV} in comparison to dSNF¹⁻⁹⁶ alone by ITC. Results are reported in new Table 1 and new Figure 8. We included a brief description of these results in the revised text (pg. 14, lines 340-349):

We also used ITC to test the contributions of dU2A' Arg20 (which contacts U14 in the loop of dU2^{SLIV}; Figure 6D, top panel) and of dU2A' C-terminal residues Gln152 and Lys153 (which contact the ribose backbone of the dU2^{SLIV} stem; Figure 7E, left panel). We generated two variants of dU2A', in which Arg20 was replaced by an alanine (dU2A'^{Arg20Ala}) or in which Gln152, Lys153 as well as the preceding Arg143, Lys149 and Lys151 were replaced by alanines (dU2A'^{mutC}). Complexes of dU2A'^{Arg20Ala}-dSNF¹⁻⁹⁶ and dU2A'^{mutC}-dSNF¹⁻⁹⁶ bound dU2^{SLIV} with only slightly reduced affinities compared to wild type dU2A' (Table 1; Figure 8E,F). While the dU2A' variants might exhibit larger effects on RNA affinity at higher temperature, the results show that under certain conditions dU2A' can enhance dU2^{SLIV}

binding by dSNF¹⁻⁹⁶ by modulation of dSNF¹⁻⁹⁶ alone and without fostering additional RNA contacts.

Minor points:

Figure S1. Please add error bars.

Done.